Manuscript prepared for Atmos. Chem. Phys.
with version 2015/04/24 7.83 Copernicus papers of the LaTeX class copernicus.cls.
Date: 6 April 2018

# The representation of solar cycle signals in stratospheric ozone. Part II: Analysis of global models

Amanda C. Maycock[1], Katja Matthes[2,3], Susann Tegtmeier[2], Hauke Schmidt[4],
Rémi Thiéblemont[5], Lon Hood[6], Hiderahu Akiyoshi[7], Slimane Bekki[5],
Makoto Deushi[8], Patrick Jöckel[9], Oliver Kirner[10], Markus Kunze[11],
Marion Marchand[12], Daniel R. Marsh[13], Martine Michou[14], David Plummer[15],
Laura E. Revell[16,17], Eugene Rozanov[16,18], Andrea Stenke[16],
Yousuke Yamashita[7,19], and Kohei Yoshida[8]

[1]School of Earth and Environment, University of Leeds, UK
[2]GEOMAR Helmholtz for Ocean Research, Kiel, Germany
[3]Christian-Albrechts Universität zu Kiel, Kiel, Germany
[4]Max Planck Institute for Meteorology, Hamburg, Germany
[5]LATMOS, Paris, France
[6]University of Arizona, Arizona, USA
[7]National Institute for Environmental Studies (NIES), Tsukuba, Japan
[8]Meteorological Research Institute, Japan Meteorological Agency, Tsukuba, Japan
[9]Deutsches Zentrum für Luft- und Raumfahrt e.V. (DLR), Institut für Physik der Atmosphäre, Oberpfaffenhofen, Germany
[10]Steinbuch Centre for Computing, Karlsruhe Institute of Technology, Karlsruhe, Germany
[11]Institut für Meteorologie, Freie Universität Berlin, Berlin, Germany
[12]Centre national de la recherche scientifique (CNRS), France
[13]National Center for Atmospheric Research, Boulder, USA
[14]CNRM UMR 3589, Météo-France/CNRS, Toulouse, France
[15]Environment and Climate Change Canada, Montréal, Canada
[16]Institute for Atmospheric and Climate Science ETH, Zurich, Switzerland
[17]Bodeker Scientific, Christchurch, New Zealand
[18]Physikalisch-Meteorologisches Observatorium, World Radiation Center, Davos, Switzerland
[19]now at: Japan Agency for Marine-Earth Science and Technology (JAMSTEC), Yokohama, Japan

*Correspondence to:* Amanda C. Maycock (a.c.maycock@leeds.ac.uk)

**Abstract.** The impact of changes in incoming solar irradiance on stratospheric ozone abundances should be included in climate simulations to aid in capturing the atmospheric response to solar cycle variability. This study presents the first systematic comparison of the representation of the 11 year solar cycle ozone response (SOR) in chemistry-climate models (CCMs) and in pre-calculated ozone databases specified in climate models that do not include chemistry, with a special focus on comparing the recommended protocols for CMIP5 and CMIP6. We analyse the SOR in eight CCMs from the SPARC/IGAC Chemistry-Climate Model Initiative (CCMI-1) and compare these with results from three ozone databases for climate models: the Bodeker Scientific ozone database, the SPARC/AC&C ozone database for CMIP5, and the SPARC/CCMI ozone database for CMIP6. The peak amplitude of the annual mean SOR in the tropical upper stratosphere (1-5 hPa) decreases by more than a factor

of two, from around 5% to 2%, between the CMIP5 and CMIP6 ozone databases. This substantial decrease can be traced to the CMIP5 ozone database being constructed from a regression model fit to satellite and ozonesonde measurements, while the CMIP6 database has been constructed from CCM simulations. The SOR in the CMIP6 ozone database therefore implicitly resembles the SOR in the CCMI-1 models. The structure in latitude of the SOR in the CMIP6 ozone database and CCMI-1 models is considerably smoother than in the CMIP5 database, which shows unrealistic sharp gradients in the SOR across the middle latitudes owing to the paucity of long-term ozone measurements in polar regions. The SORs in the CMIP6 ozone database and the CCMI-1 models show a seasonal dependence with enhanced meridional gradients at mid to high latitudes in the winter hemisphere. The CMIP5 ozone database does not account for seasonal variations in the SOR, which is unrealistic. Sensitivity experiments with a global atmospheric model without chemistry (ECHAM6.3) are performed to assess the atmospheric impacts of changes in the representation of the SOR and solar spectral irradiance (SSI) forcing between CMIP5 and CMIP6. The larger amplitude of the SOR in the CMIP5 ozone database compared to CMIP6 causes a likely overestimation of the modelled tropical stratospheric temperature response between 11 year solar cycle minimum and maximum by up to 0.55 K, or around 80% of the total amplitude. This effect is substantially larger than the change in temperature response due to differences in SSI forcing between CMIP5 and CMIP6. The results emphasise the importance of adequately representing the SOR in global models to capture the impact of the 11 year solar cycle on the atmosphere. Since a number of limitations in the representation of the SOR in the CMIP5 ozone database have been identified, we recommend that CMIP6 models without chemistry use the CMIP6 ozone database and the CMIP6 SSI dataset to better capture the climate impacts of solar variability. The CMIP6 solar-ozone coefficients are published with this manuscript.

## 1 Introduction

Stratospheric heating rates are enhanced between the minimum and maximum phases of the approximately 11 year solar cycle through two main effects: (1) absorption of enhanced incoming ultraviolet (UV) radiation; and (2) enhanced ozone concentrations (brought about by increased photochemical production) (e.g. Penner and Chang (1978); Brasseur and Simon (1981)). These radiative changes can drive feedbacks onto stratospheric dynamics, leading to amplified signals of solar cycle variability in regional surface climate via stratosphere-troposphere dynamical coupling (e.g. Kuroda and Kodera (2002)). To understand and model the impacts of solar cycle variability on the atmosphere it is therefore necessary to account for the characteristics of solar spectral irradiance (SSI) variability and the associated solar cycle ozone response (SOR) (e.g. Haigh (1994)).

In Part I of this study, Maycock et al. (2016) examined the SOR in a number of recently updated and merged satellite ozone datasets from the instruments SAGE II, GOMOS, OSIRIS and SBUV. The present Part II focuses on the representation of the SOR in global climate and chemistry-climate

models. At a minimum, models must include a sufficiently detailed representation of SSI and the SOR to realistically simulate solar cycle impacts on the atmosphere. The global models routinely employed in international scientific assessments (e.g. IPCC, WMO Ozone Assessments) typically represent atmospheric ozone in one of two ways. Chemistry-climate models (CCMs) include inter-active stratospheric chemistry and explicitly simulate a SOR that is consistent with their photolysis, radiation and transport schemes provided that SSI variations are adequately (i.e. with sufficiently high spectral resolution) represented. A small but growing number of CCMs also include the chemical effects of galactic cosmic rays and solar energetic particles, though these effects are not explicitly considered in this study. Conversely, global climate models do not routinely include interactive chemistry and must therefore prescribe a pre-calculated ozone distribution to the radiation scheme, which is usually taken from observations and/or chemical models. Thus, if climate models without chemistry are to capture the full atmospheric response to solar cycle variability, they must prescribe an ozone dataset that includes a representation of the SOR.

Understanding and constraining the SOR is a long-standing scientific issue and numerous studies have analysed its representation in observations (see Maycock et al. (2016) and references therein) and CCMs (e.g. Austin et al. (2008); Sekiyama et al. (2006); Lee and Smith (2003); Egorova et al. (2014); Dhomse et al. (2011, 2016); Hood et al. (2015); SPARC CCMVal (2010)). Older generations of CCMs (e.g. CCMVal-1/2) showed a positive annual mean SOR of up to $\sim$2.5% peaking in the tropics between $\sim$3-5 hPa and a maximum tropical mean temperature response in the upper stratosphere of $\sim$0.5-1.1 K (Austin et al., 2008; SPARC CCMVal, 2010). Since these earlier studies the CCMs from different centres have been significantly revised and developed (e.g. Morgenstern et al. (2017)). The more recent study by Hood et al. (2015) only analysed a small number of CCMs that participated in the World Climate Research Programme fifth Coupled Model Intercomparison Project (CMIP5). Thus there has been no detailed comparison of the SOR in a larger set of CCMs since Austin et al. (2008). Furthermore, Hood et al. (2015) compared the few CMIP5 models with older versions of satellite datasets that have since been updated and extended leading to pronounced changes in their representation of the SOR (Maycock et al., 2016; Dhomse et al., 2016). Hence one goal of this study is to provide an important update by evaluating the SOR in the latest models from the SPARC/IGAC Chemistry Climate Model Initiative (CCMI-1) and comparing them to some of the recently updated and extended satellite datasets discussed in Part I.

A further motivation for this study is the recent analysis of the climate response to the solar cycle in CMIP5, which included models with and without interactive stratospheric chemistry. The CMIP5 models showed a large spread ($\sim$0.3-1.2 K) in the peak amplitude of the tropical stratospheric temperature response between the minimum and maximum phases of the 11 year solar cycle (Mitchell et al., 2015). This spread may be due to differences in the prescription of SSI, in the accuracy of model radiative transfer schemes (Nissen et al., 2007; Forster et al., 2011), and/or in the representation of the SOR. However, the quantitative importance of any one of these factors is unclear. All

CMIP5 models were recommended to use the Naval Research Laboratory Spectral Solar Irradiance 1 (NRLSSI-1) dataset (Wang et al., 2005). Those without chemistry were further recommended to pre-

scribe ozone from the SPARC/Atmospheric Chemistry and Climate (AC&C; www.igacproject.org) ozone database (Cionni et al. (2011); hereafter referred to as CMIP5 ozone database). The historical part of this ozone database was largely based on multiple regression model fit to satellite observations (see Section 2.1.2). It is therefore plausible that differences in the representation of SOR made an important contribution to the spread in atmospheric thermal and dynamical responses to the solar

cycle in CMIP5 models; we investigate this hypothesis further in this study.

As was the case in CMIP5, CMIP6 will include a mix of models with and without explicit stratospheric chemistry. A new ozone database has been created for CMIP6 models without chemistry (hereafter referred to as CMIP6 ozone database; see https://esgf-node.llnl.gov/projects/input4mips). It is therefore important to document the SOR in the new CMIP6 ozone database and compare it to

the previous CMIP5 database, since these fields are routinely deployed in climate models and differences may lead to changes in the modelled responses to solar forcing between CMIP5 and CMIP6. In addition to documenting the SOR in the CMIP6 ozone database, this study is published with the ozone coefficients derived from the analysis (https://doi.org/10.5518/348), so they can be used in other modelling projects (e.g. Jungclaus et al. (2017)).

Another factor to consider for modelling solar cycle effects on the atmosphere is the representation of the annual cycle in the SOR (Soukharev and Hood, 2006; Hood et al., 2015; Maycock et al., 2016). Hood et al. (2015) found that the three CMIP5 CCMs that simulated large horizontal gradients in the SOR in the upper stratosphere in early winter also showed Northern hemisphere high latitude dynamical responses over the 11 year solar cycle that compared more closely with reanalysis data.

The enhancement of the SOR at high latitudes is related to coupling between chemistry and transport processes for ozone and may play a role in driving the 'top-down' mechanism for the solar cycle influence on high latitude regional surface climate (see e.g. Gray et al. (2010)). It is therefore also important to compare the representation of the annual cycle in the SOR in current CCMs and in the pre-calculated ozone databases used in climate models.

The objectives of this study are therefore:

 – to provide an update to Austin et al. (2008) by analysing the SOR in CCMI-1 models.

 – to document the SOR in the new CMIP6 ozone database and compare this to previous pre-calculated ozone databases including CMIP5.

 – to compare the SOR in CCMs and ozone databases with recently updated and extended satellite records.

 – to perform atmospheric model experiments to quantify the impact of differences in the SOR between CMIP5 and CMIP6 on the simulated atmospheric response to the 11 year solar cycle.

Collectively these objectives provide a comprehensive assessment of the represention of the SOR in current CCMs and global climate models. The outline of the remainder of the paper is as follows: Section 2 describes the data and methods used to analyse the SOR, Section 3 presents the results, and Section 4 summarises our findings.

## 2 Methods

### 2.1 Models and ozone datasets

#### 2.1.1 The CCMI-1 models

Data are analysed from eight CCMI-1 models downloaded from the British Atmospheric Data Centre (Hegglin and Lamarque, 2015). The models analysed are: CCSRNIES-MIROC3.2, CESM1(WACCM), CMAM, CNRM-CM5-3, EMAC(L90), LMDz-REPROBUS-CM5 (L39), MRI-ESM1r1, and SO-COL3 (see Table 1). These models include the minimum requirements for capturing the SOR (i.e. a prescription of SSI variability in the chemistry scheme). A detailed description of the models is given by Morgenstern et al. (2017).

Data are analysed from the REF-C1 simulations, which include observed time-varying sea surface temperatures (SST) and sea ice, well-mixed greenhouse gases, volcanic aerosols, and SSI forcing from NRLSSI-1 (Eyring et al., 2013b). Thus, in contrast to the coupled atmosphere-ocean CMIP5 models analysed by Hood et al. (2015), the CCMI-1 REF-C1 simulations do not include an interactive ocean. The REF-C1 simulations start in January 1960 but terminate in different years for each model, so for consistency we analyse the 50 year period 1960-2009 which is common to all the simulations. All available ensemble members are analysed for each model (see Table 1).

The representation of the QBO differs across the CCMI-1 models (Morgenstern et al., 2017). Some of the models simulate a spontaneous QBO (MRI-ESM1r1, EMAC(L90)), some models include a QBO by nudging tropical stratospheric zonal winds towards observations (CCSRNIES-MIROC3.2, CESM1(WACCM), SOCOL3), and some include no representation of the QBO (CMAM, CNRM-CM5-3, LMDz-REPROBUS-CM5). In EMAC(L90) a weak nudging towards the observed QBO with a relaxation timescale of 58 days is applied to ensure the same phasing as the observed QBO, whereas in CCSRNIES-MIROC3.2, CESM1(WACCM) and SOCOL3 the QBO is nudged more strongly (5-10 day timescale). For those models that include QBO variability, two additional orthogonal QBO indices are included in the multiple linear regression (MLR) model calculated from the modelled zonal mean zonal wind fields (see Section 2.2).

#### 2.1.2 The CMIP5 ozone database

The CMIP5 ozone database consists of monthly mean ozone mixing ratios on 24 pressure levels spanning 1000-1 hPa for the period 1850-2100 (Cionni et al., 2011). Data are provided on a regular

$5 \times 5°$ longitude/latitude grid. Ozone values are provided as a 2-D (i.e. zonal mean) field in the stratosphere (at pressures less than 300 hPa) and as a 3-D field in the troposphere, with a blending across the tropopause. The tropospheric part of the database was constructed from CCM simulations. For the stratosphere, the historical portion of the database (1850-2009) was constructed from observations using an MLR model (that includes solar variability as one of the independent variables) fit to SAGE I and SAGE II version 6.2 satellite data and polar ozonesondes following the method of Randel and Wu (2007). A SOR is therefore implicitly included in the historical portion of the CMIP5 ozone database that will resemble the observations input to the MLR model. However, owing to the paucity of long-term ozone measurements at high latitudes, the SOR was only included between $\pm 60°$ latitude. This limitation led some CMIP5 modelling groups to make alterations to the CMIP5 ozone database, including extrapolation of the SOR coefficients at $\pm 50°$ latitude to the poles using a cosine latitude weighting. The CMIP5 models known to have employed this 'Extended CMIP5 ozone database' include HadGEM2-CC (Osprey et al., 2013), MPI-ESM (Schmidt et al., 2013) and CMCC-CC (Cagnazzo, 2016, pers. comms.). Note that the historical portion of the CMIP5 ozone database did not include a representation of QBO variability in ozone.

The future portion of the CMIP5 ozone database for the stratosphere was based on CCM simulations from CCMVal-2 (Cionni et al., 2011). However, owing to uncertainties in how individual CMIP5 models would represent SSI variations over the 21st century, the future portion of the CMIP5 ozone database did not include a SOR. For consistency, a SOR was thus added to the future period in the Extended CMIP5 ozone database using regression coefficients for the SOR derived from the historical period (Schmidt et al. (2013); Osprey et al. (2013); C. Cagnazzo, 2016, pers. comms.).

The CMIP5 ozone database is described in full by Cionni et al. (2011) and is available at the time of writing from: https://cmip.llnl.gov/cmip5/forcing.html#ozone_forcing. A description of the CMIP5 models that employed the CMIP5 ozone database is given by Eyring et al. (2013a).

### 2.1.3 The CMIP6 ozone database

The CMIP6 ozone database for the historical period (1850-2014) consists of monthly mean ozone mixing ratios on 66 pressure levels spanning 1000-0.0001 hPa. Data are provided as a 3-D field on a regular $2.5 \times 1.9°$ longitude/latitude grid. The database is constructed using a weighted average of simulations from two CCMs (CESM1(WACCM) and CMAM) (M. Hegglin, pers. comms.). The CMIP6 ozone database was downloaded from: https://esgf-node.llnl.gov/projects/input4mips.

The simulations from the two constituent CCMs include prescribed SSTs, sea ice, well-mixed greenhouse gas concentrations and aerosols. Surface emissions of $NO_x$ and other tropospheric ozone precursor gases are also prescribed. Both CCMs represent SSI variability in their radiation and chemical schemes. However, only CESM1(WACCM) includes the chemical effects of energetic particle precipitation.

There are some differences in the set-up of the CCM simulations used to create the CMIP6 ozone database compared to the CCMI-1 versions of the same models (see Section 2.1.1), which may affect the representation of the SOR. The version of CMAM for the CMIP6 ozone database used historical stratospheric aerosols and solar variability, similar to in REF-C1, extended back to 1850. However, SSTs and sea-ice were prescribed from a CanESM2 historical simulation performed for CMIP5 rather than from observations. The temporal variability in SSI for CMAM was taken from the CMIP6 SSI dataset (Matthes et al., 2017), but the variations were applied to the long-term background spectrum from NRLSSI-1. This is in slight contrast to the CCMI-1 version of CMAM that used both SSI variability and the background spectrum from NRLSSI-1. However, Matthes et al. (2017) showed that the slightly weaker variability over the solar cycle at shorter UV wavelengths in NRLSSI-1 only reduced the amplitude of the tropical mean SOR in a CCM by $\sim$0.3% compared to a reference of $\sim$2%. This difference is therefore likely to have only a small effect on the SOR in the configurations of CMAM implemented for CCMI-1 and the CMIP6 ozone database. Neither CMAM simulation includes nudging of the QBO.

There are also some differences in the configuration of CESM1(WACCM) used for the CMIP6 ozone database compared to CCMI-1. The CESM1(WACCM) CCMI-1 runs prescribed the NRLSSI-1 data at daily resolution, whereas the version for the CMIP6 ozone database used annual values as these extend back to 1850. In the lower thermosphere, values of the F10.7cm flux and Kp index used to parametrize the chemical effects of energetic particle precipitation were taken from observations in CCMI-1 and from a proxy record in the simulation for the CMIP6 ozone database. Furthermore, the simulation for the CMIP6 ozone database did not include solar proton events or galactic cosmic ray effects. Both versions of CESM1(WACCM) used observed SSTs and include a nudged QBO towards observed tropical winds. In summary, there are some differences in the experimental set-ups of the two CCMs used to create the CMIP6 ozone database, in particular that they use slightly different representations of SSI variability, they do not both include QBO variability and they use different SST datasets.

### 2.1.4 The Bodeker ozone database

Bodeker et al. (2013) describe a new observation based ozone database for climate models covering the period 1979-2007. Monthly and zonal mean ozone mixing ratios are provided on 70 pressure levels spanning 878-0.05 hPa on a regular 5° latitude grid. The ozone field is constructed from a large number of satellite and ozonesonde observations from the Binary DataBase of Profiles (BDBP; Hassler et al. (2008)) used to fit an MLR model that includes terms for equivalent effective stratospheric chlorine (EESC), a linear trend, the QBO, the El Niño Southern Oscillation (ENSO), the solar cycle, and the Mt Pinatubo volcanic eruption. We note that since the BDBP contains SAGE II v6.2 mixing ratio data, this is likely to provide a strong constraint on the SOR in the tropics and subtropics. See Maycock et al. (2016) and Dhomse et al. (2016) for a discussion

of the differences in the SOR in SAGE II v6.2 and v7.0 data. To generate a spatially complete ozone field a single MLR fit is performed for all points on a given pressure surface to enable regression coefficients to be derived for latitudes where the observations are relatively sparse (e.g. in polar regions). We use the Tier 1.4 product from the Bodeker ozone database, which includes contributions from all the MLR basis functions. The Bodeker ozone database was downloaded from http://www.bodekerscientific.com/data/monthly-mean-global-vertically-resolved-ozone.

## 2.2 The multiple linear regression (MLR) model

Multiple linear regression models have been used to analyse drivers of secular trends and variability in stratospheric ozone for many decades (see e.g. Staehelin et al. (2001) and references therein). In the context of extracting the SOR from ozone timeseries, there is a long history of similar methods being applied to both satellite observations (e.g., Soukharev and Hood (2006); Remsberg (2008); Tourpali et al. (2007); Remsberg and Lingenfelser (2010); Dhomse et al. (2016); Lee and Smith (2003); Lean (2014); Randel and Wu (2007); Merkel et al. (2011); Maycock et al. (2016) and chemistry-climate models (Austin et al., 2008; Sekiyama et al., 2006; Lee and Smith, 2003; Egorova et al., 2014; Dhomse et al., 2011, 2016; Hood et al., 2015; SPARC CCMVal, 2010). Here we follow the methodology described by Maycock et al. (2016), which is very similar to the methods described in those earlier studies. Briefly, the zonal mean ozone data are deseasonalised by removing the long-term monthly mean at each latitude and pressure level. As in past studies, we then perform an MLR analysis on the timeseries of monthly mean anomalies at each location, $O_3'(t)$, to diagnose the solar cycle component:

$$
\begin{aligned}
O_3'(t) = A \times F10.7(t) + B \times CO_2(t) + C \times EESC(t) \\
+ D \times ENSO(t) + E \times QBO_A(t) \\
+ F \times QBO_B(t) + r(t),
\end{aligned}
\tag{1}
$$

where *r(t)* is a residual. The annual-mean SOR is calculated by regressing all months as a single timeseries. The monthly SOR is calculated by regressing timeseries of year-to-year anomalies for individual months. The monthly mean basis functions in Equation 1 are the F10.7cm radio solar flux, the $CO_2$ concentration at Mauna Loa, the equivalent effective stratospheric chlorine (EESC), and the Nino 3.4 index to represent ENSO. The F10.7cm flux is used to represent solar activity because it has been shown to be well correlated with indices for UV radiation (e.g. Floyd et al. (2005)), the key driver of the stratospheric ozone response. The results presented in Section 3 assume a difference of 130 solar flux units (1 SFU = $10^{-22}$ Wm$^{-2}$Hz$^{-1}$) as a representative amplitude of the 11 year solar cycle. The Nino 3.4 index is computed as the standardised mean SST averaged over the region 5°S–5°N and 120°W–170°W. For those CCMI-1 models and ozone databases that include QBO variability (see Table 1), the QBO indices are calculated as the first two principal component timeseries of the tropical ($\pm 10°$, 5-70 hPa) zonal mean zonal winds. The ozone response

to volcanic aerosols is non-linear through time owing to changing levels of inorganic chlorine in the atmosphere (Tie and Brasseur, 1995). Thus, rather than including a term in the MLR model to represent volcanic effects on ozone, data from the 2 year periods following the three major tropical volcanic eruptions since 1960 are excluded from the analysis: Mount Agung (February 1963), El Chiĉhon (March 1982) and Mount Pinatubo (June 1991). Figure 1 shows example timeseries of the MLR basis functions from 1960-2009 in arbitrary units. In this example the ENSO and QBO indices are based on observations. The coefficients A–F in Equation 1 are calculated using linear least squares regression.

One important issue for MLR analysis is the handling of possible autocorrelation in the regression residuals, $r(t)$, and the effect on the estimation of statistical uncertainty in the results. A Durbin-Watson test reveals significant serial correlation in the regression residuals in many locations for lags of one and two months, particularly in the middle and polar lower stratosphere. Such serial correlation can lead to spurious overestimation of the statistical significance of the regression coefficients and we therefore include an autoregressive term in the regression model. Given the significant serial correlations in some regions up to a lag of two months, a second order autoregressive noise process (AR2) is used, which assumes the residuals $r(t)$ have the form:

$$r(t) = ar(t-1) + br(t-2) + w(t), \tag{2}$$

where $a$ and $b$ are constants and $w(t)$ is a white noise process. This is identical to the approach employed in Maycock et al. (2016) and the recent SPARC SI$^2$N analysis of ozone trends (Tummon et al., 2015; Harris et al., 2015). The autocorrelation term is not included in the analysis of the monthly SOR because the residuals are approximately uncorrelated from year-to-year. Unless otherwise stated, the statistical significance of the SOR extracted using the MLR model is assessed using a two-tailed Student's t-test with a null hypothesis that the magnitude of the SOR is indistinguishable from zero. We apply a threshold to determine whether the null hypothesis can be rejected at a 95% confidence level.

It is a challenge in geophysical science to develop statistical methods to extract forced signals from complex timeseries. The implementation of multiple linear regression analysis as described above may have a number of limitations, including (but not limited to): assumption that the input basis functions have zero uncertainty; difficulties in separating a signal from noise in relatively short or sparse records (Damadeo et al., 2014); and issues arising from degeneracy between basis functions (Chiodo et al., 2014). We have not attempted to account for these factors in the results shown in Section 3.

## 2.3 Atmospheric model sensitivity experiments

To explore the atmospheric impacts of different representations of the SOR, simulations were carried out with the atmospheric general circulation model ECHAM6.3, which is an update of the

ECHAM6.1 model (Stevens et al., 2013) used as atmospheric component of the Max Planck Institute Earth System Model (Giorgetta et al., 2013) in CMIP5 simulations. Model changes from version 6.1 to 6.3 are mainly related to fixes of bugs described by Stevens et al. (2013), efforts to ensure energy conservations, an update of the radiation scheme, which is now the PSrad (Pincus and Stevens, 2013) version of the RRTMG code (Iacono et al., 2008), and retuning. If the same forcings are used, temperature effects of solar cycle variability in ECHAM6.3 compare well to those described for ECHAM6.1 (Schmidt et al., 2013). The model experiments performed here use a horizontal resolution of T63 ($\sim$140 $\times$ 210 km) with 47 vertical levels up to a lid of 80 km.

It is known that the ECHAM6.3 radiation code does not cover wavelengths below 200 nm and therefore the important Schumann-Runge bands and Lyman-alpha lines of ozone are not captured (Sukhodolov et al., 2014). This results in a too weak radiative response to the imposed solar forcing particularly in the mesosphere. Therefore we focus our analysis on the stratospheric response where most of the absorption occurs at higher UV wavelengths, and the performance of ECHAM6.3 is comparable to models with a more comprehensive radiative code (Sukhodolov et al., 2014).

We have performed five time-slice simulations with ECHAM6.3 each lasting for 50 years. The control simulation uses average boundary conditions as specified for the CMIP5 AMIP simulation, i.e. for all boundary conditions such as SSTs, greenhouse gas concentrations, SSI and prescribed atmospheric ozone we have used multi-year averages of the CMIP5 recommended values for the years 1978 to 2008. Four sensitivity simulations have then been performed in which solar maximum minus solar minimum differences in either atmospheric ozone concentrations or both ozone and SSI have been added to the respective fields of the control simulation. For solar maximum and minimum conditions we have used average values over the years 1985-1986 and 1981-1982, respectively. According to the solar irradiance recommendations for CMIP6 (Matthes et al., 2017) these are characterized by a difference of 126.1 SFU, and are therefore closely comparable to the results presented for the SOR, which assume a representative solar cycle amplitude of 130 SFU. Ozone anomalies were either calculated from the respective years of the Extended CMIP5 ozone database (Schmidt et al. (2013)) or using the monthly SOR coefficients from the CMIP6 ozone database shown in Section 3.3. The corresponding SSI anomalies are either calculated from the CMIP5 recommended NRLSSI-1 dataset (Wang et al., 2005) or from the CMIP6 recommended solar forcing dataset (Matthes et al., 2017).

## 3 Results

### 3.1 The SOR in CCMI-1 models

Figure 2 shows timeseries of deseasonalised tropical (30°S-30°N) and monthly mean percent ozone anomalies at select pressure levels (1, 3, 5, 10, 30 hPa) for the eight CCMI-1 models described in Section 2.1.1. Anomalies are defined relative to the period 1985-2009. Also plotted in Figure

2 are timeseries from two satellite datasets discussed in Part I of this study: SBUVMOD VN8.6 (Frith et al., 2014) (black dashed) and SAGE-GOMOS 1 (Kyrölä et al., 2015) (black solid). For completeness, the timeseries of absolute ozone mixing ratios from the models are shown in the Supplementary Material (Figure S1).

The CCMI-1 models show a long-term decline in stratospheric ozone abundances, particularly in the mid and upper stratosphere. This is the result of increasing atmospheric loading of inorganic chlorine and bromine over this period and is consistent with results from earlier CCM studies (e.g. Eyring et al. (2006); SPARC CCMVal (2010)). At $1\,hPa$, the trend in ozone between 1979-1997 computed by linear regression ranges from -1.9 to -2.6 % decade$^{-1}$ across the models. At $3\,hPa$, the range in trends is -4.1 to -5.1 % decade$^{-1}$. These values are within the uncertainty bounds of satellite observed ozone trends over this period (Harris et al., 2015).

In addition to a long-term decline, Figure 2 shows quasi-decadal variations in ozone in the upper stratosphere that are approximately in phase with the 11 year solar cycle. These are a marker of the SOR which is evident in the raw ozone timeseries and can be seen as a peak around the decadal timescale in the modelled ozone power spectra (see Supplementary Material Figure S2). There is larger interannual and multi-year variability in ozone at 10 and $30\,hPa$ where some models show enhanced variability associated with the QBO. The modelled evolution of the tropical ozone anomalies is generally in good agreement with the observation data in Figure 2, with some exceptions where the satellite records show larger amplitude short-term fluctuations that may be related to incomplete spatial and temporal sampling.

Figure 3 shows latitude-pressure cross-sections of the annual mean SOR in the eight CCMI-1 models (Figure 3(a-h)) along with the multi-model mean (Figure 3(i)). For the individual models, the statistical significance of the SOR is assessed using a two-tailed Student's t-test with a threshold for rejecting the null hypothesis at the 95% confidence level (see Section 2.2). The robustness of the CCMI-1 multi-model mean SOR is assessed by distinguishing regions where the magnitude of the SOR is greater than $\pm 2$ s.d. of the intermodel spread. Figure 3 can be compared with Figure 1 in Austin et al. (2008) and Figure 1 in Hood et al. (2015) which show similar plots for the CCMVal-1 and CMIP5 models, respectively.

All of the CCMI-1 models analysed show a significant positive SOR of up to $\sim$2% between 1-$10\,hPa$. This is less than half the peak amplitude of the SOR in the SAGE II v6.2 mixing ratio dataset and is more comparable to the SOR amplitude in SAGE II v7.0 mixing ratios and the SBUVMOD VN8.6 dataset (see Figures 4 and 12 in Maycock et al. (2016)). The results from the CCMI-1 models are broadly consistent with the results from CCMVal-1 models (Austin et al., 2008). The main exception is the absence in the multi-model mean of a strong SOR in the tropical lower stratosphere. An enhanced SOR in the tropical lower stratosphere has been identified in satellite observations, albeit with large uncertainties (Gray et al., 2009; Austin et al., 2008; Soukharev and Hood, 2006; Maycock et al., 2016), and it has been postulated this may be associated with a change in the strength

of the Brewer Dobson circulation. The CCMVal-1 multi-model mean showed a SOR of around 5% per 130 SFU at ∼50 hPa (see Figure 4(d) in Austin et al. (2008)), as compared to around 1% in the CCMI-1 multi-model mean (Figure 3(i)). However, there was large intermodel spread in this signal
across the CCMVal-1 models and the multi-model mean SOR was dominated by strong responses in a few models that only ran for a short period (1980-2004) during which aliasing effects with other climatic factors can be significant (Chiodo et al., 2014). Since the analysis shown here extends for a longer period and excludes the post-volcanic epochs, this is a plausible reason for the apparent difference in the SOR in the tropical lower stratospheric between the CCMI-1 and CCMVal-1 mod-
els. One of the CCMI-1 models (SOCOL3) appears to show an enhanced SOR in the tropical lower stratosphere, which is similar in amplitude to that seen in some CCMVal-1 models. However, this feature shows some sensitivity to the choice of autoregressive model in the MLR model probably because the decorrelation timescale for the regression residuals in the tropical lower stratosphere is longer than two months in SOCOL3 and some of the other CCMs (not shown). Further analysis of
the Transformed Eulerian Mean residual vertical velocity does not reveal a substantial change in the rate of upwelling in the tropical lower stratosphere in any of the models (not shown).

The month-by-month SORs in the individual CCMI-1 models (see Supplementary Material Figures S3-10) show a significant positive SOR in the tropical upper stratosphere throughout the year, but enhanced SOR amplitudes at high latitudes particularly in the winter and spring seasons. This
behaviour, which is also seen in some satellite ozone datasets (e.g. Maycock et al. (2016)), cannot be understood from photochemical processes alone and must therefore be related to stratospheric circulation changes (e.g. Kuroda and Kodera (2002)). Such localised changes in ozone will be associated with a radiative perturbation that could lead to feedbacks onto circulation (Hood et al., 2015), and thus it may be important to account for this seasonal variation in the SOR in model simulations.

**3.2   The SOR in ozone databases for climate models**

Figure 4 shows timeseries from 1960-2011 of deseasonalised tropical and monthly mean fractional ozone anomalies at select stratospheric levels (1 to 30 hPa) from the Bodeker (orange line), CMIP5 (red) and CMIP6 (blue) ozone databases. Also plotted in black are the same satellite datasets as shown in Figure 2. Anomalies are defined relative to the period 1985-2007. The Extended CMIP5
ozone database is not shown because it is identical to original CMIP5 database in the tropics.

Although the timeseries have been deseasonalised, the CMIP5 and Bodeker ozone databases show a residual annual cycle particularly in the upper stratosphere. This is because in these databases the amplitude of the ozone annual cycle is larger in the early part of the record, when the background levels are higher, and smaller in the latter part of the record following the long-term decline in ozone.
Since the ozone anomalies in Figure 4 are shown as anomalies from the 1985-2007 mean, there is therefore a residual annual cycle particularly in the period before 1985. Conversely, the CMIP6

database, which is constructed from CCM simulations, does not show a significant change in the amplitude of the ozone annual cycle over time.

At 1 hPa, the CMIP5 and Bodeker databases show a larger linear trend in ozone over 1979-2007
(diagnosed using linear regression) of around -3.5 % decade$^{-1}$ compared to -1 % decade$^{-1}$ in the CMIP6 database. The latter is, as expected, similar to the long-term ozone trends in the CCMI-1 models shown in Figure 2. At 3 hPa, the CMIP5 database also shows a larger long-term decrease in ozone by around a factor of two compared to the Bodeker and CMIP6 databases. Thus, a model that uses the recommended CMIP6 ozone database might be expected to show weaker upper stratospheric
cooling over recent decades compared to an equivalent simulation using the CMIP5 database, owing to the smaller negative trend in upper stratospheric ozone.

At 10 and 30 hPa, the Bodeker and CMIP6 databases show a QBO signal in ozone, whereas the CMIP5 database does not include QBO variability. This is an important distinction because a model that employs the CMIP6 ozone database, but which does not simulate a dynamical QBO, will impose
a QBO-ozone signal that may alter the model's behaviour. Alternatively, a model that internally generates a dynamical QBO that is not in phase with the prescribed QBO-ozone signal in the CMIP6 ozone database will be subject to a diabatic heating anomaly from ozone that is inconsistent with the model's dynamical evolution. Both of these cases would be physically unrealistic. However, a model that nudges a QBO towards observations and uses the CMIP6 ozone database should have
a more consistent representation of temporal variability in winds and ozone associated with the QBO. Modelling groups may therefore choose to post-process the CMIP6 ozone database in order to treat the QBO-ozone signal in a consistent manner for their model. Note that since the CMIP6 ozone database is produced by averaging two CCMs, one that does include QBO-ozone variability (CESM1(WACCM)) and one that does not (CMAM), the QBO-ozone signal is weaker in the CMIP6
ozone database than in the CESM1(WACCM) model alone (compare blue line in Figure 2 with dark pink line in Figure 4). The absence of a QBO-ozone signal in the CMIP5 ozone database means that CMIP5 models that simulated a QBO would have neglected any radiative QBO feedback from ozone.

Figure 5 shows latitude-pressure cross-sections of the annual mean SOR in the three ozone databases
shown in Figure 4 and the Extended CMIP5 ozone database. In the tropics, the Bodeker ozone database, Figure 5(a), shows a positive SOR of up to 4% peaking at around 2-3 hPa with a distinct minimum at ~10 hPa. The latitudinal structure of the SOR is smoother than in the SAGE II v6.2 mixing ratio data (cf. Figure 4(d) of Part I) probably because the construction of the Bodeker database fits an MLR model to all data points along pressure surfaces rather than to individual latitude bands.
At high latitudes, the magnitude of the SOR in the Bodeker database is small and the spatial structure is noisy likely because of the small number of observations there. In the lower stratosphere, the Bodeker database indicates a positive SOR at most latitudes, as was found in a number of satellite

ozone datasets in Part I. However, the uncertainty in the magnitude of the SOR at these levels is comparatively large (see below).

The SOR in the CMIP5 ozone database, Figure 5(b), shows a very similar structure to that found in SAGE v6.2 mixing ratios (cf. Figure 4(d) in Part I), consistent with those data forming the backbone for the historical portion of the dataset (Cionni et al., 2011). Note that the MLR fits were applied separately at each latitude band in the construction of the CMIP5 database, and this likely explains why the horizontal structure of the SOR is more heterogeneous than in the Bodeker ozone database.

In particular the three peaked structure of the SOR found in the tropical upper stratosphere in the SAGE II v6.2 mixing ratio dataset is evident in the CMIP5 ozone database. The sharp cut-offs in the SOR at $\pm 60°$ latitude are spurious and result from a lack of data points to define a SOR at high latitudes. As described in Section 2.1.2, the Extended CMIP5 ozone database, Figure 5(c), applied a simple extrapolation to introduce a SOR in the extratropics. The details of this structure, which shows

a positive SOR extending into the northern extratropics and in the southern hemisphere a negative SOR at pressures greater than $\sim 5$ hPa poleward of $60°$S, is likely to be subject to considerable uncertainties owing to the simple spatial filling method employed.

In the CMIP6 ozone database, Figure 5(d), the amplitude of the SOR is around 1-2% in the upper stratosphere, which is as expected broadly consistent with the CCMI-1 results shown in Figure 3. The

peak amplitude of the SOR is therefore 2-3 times smaller, and is considerably smoother in latitude, than in the CMIP5 ozone database. In the lowermost tropical stratosphere, the CMIP6 database shows a positive SOR of up to $\sim 3\%$ in the southern tropics. This is slightly larger than the SOR in the tropical lower stratosphere simulated by the CCMI-1 versions of the two CCMs used to produce the CMIP6 ozone database (CESM1(WACCM) and CMAM) (see Figure 3(b-c)). To further investigate

the vertical structure of the tropical SOR and its uncertainties, Figure 6 shows the best estimate tropical ($30°$S-$30°$N) mean SOR along with the 2.5-97.5% confidence intervals for the climate model ozone databases and the two satellite datasets from Figure 2 (see Part I). Also shown in grey shading is the range of the best estimate SORs from the eight CCMI-1 models. The best estimate SOR in the tropical lower stratosphere ranges from a small negative signal in the CMIP5 ozone database to

+6% in the Bodeker ozone database. In the CMIP6 ozone database, the best estimate tropical SOR is 2% at 80 hPa, which is, as expected, within the range of the signals in the CCMI-1 models. The substantial spread amongst the estimates along with the large uncertainties reinforces the challenge of constraining the SOR in the tropical lower stratosphere (e.g. Marsh and Garcia (2007)). Despite the relatively large uncertainties, the SOR in the tropical lower stratosphere is larger in the CMIP6

database compared to in CMIP5; this may be important for the modelled atmospheric response to solar variability in CMIP5 and CMIP6 models (see Section 3.4). Figure 6 further confirms that the two climate model ozone databases that include SAGE II v6.2 mixing ratio data (CMIP5 and Bodeker), show a significantly stronger SOR in the tropical upper stratosphere. This is likely to be unrealistically large since the updated SAGE II v7.0 mixing ratio data, which show a smaller SOR

in the tropical upper stratosphere (Maycock et al., 2016), exhibit a more realistic representation of the relationship between upper stratospheric ozone and temperature compared to SAGE II v6.2 data (Dhomse et al., 2016).

### 3.3 Comparison of SOR annual cycle in CMIP5 and CMIP6 ozone databases

Earlier studies have shown evidence for an annual cycle in the structure and amplitude of the SOR
in satellite observations (e.g. Soukharev and Hood (2006); Maycock et al. (2016)). Figure 7 shows the monthly mean SOR in the Extended CMIP5 ozone database and Figure 8 shows the same for the CMIP6 ozone database. The SOR in the CMIP5 database has a fixed structure and constant amplitude in all months; the small annual cycle in the fractional SOR amplitude arises purely from the annual cycle in background ozone concentrations. There are well understood photochemical arguments for
why the structure of the SOR is expected to track the position of the Sun through the year (Haigh, 1994). Furthermore, the coupling between ozone and stratospheric dynamics may lead to variations in the SOR at high latitudes in some months due to the formation in winter of the polar vortices and their subsequent break-up in spring (Hood et al., 2015). For these reasons a complete absence of seasonal variation in the SOR as found in the CMIP5 ozone database is unrealistic. In contrast, the
SOR in the CMIP6 ozone database, Figure 8, shows greater seasonal variation. Locally enhanced signals in the SOR are found in the high latitudes in winter and spring, which may be linked to variations in the strength of the polar vortex (Kuroda and Kodera, 2002). Thus, including some semblence of an annual cycle in the SOR, as seen in Figure 8, is likely to be a truer reflection of the behaviour of the real atmosphere than the complete absence of an SOR annual cycle as in Figure 7.
However, the associated uncertainties in the monthly SORs are larger compared to the annual mean results presented in the previous section, and there are quantitative differences between the SOR annual cycle in the CMIP6 ozone database and that estimated from satellite observations (cf. e.g. Figure 13 of Maycock et al. (2016)). Such differences may result from uncertainties in estimating the SOR from relatively short observational records, from errors in the representation of the SOR
in the models used to construct the CMIP6 ozone database, or a combination of factors. Thus we should not consider the evolution of the monthly SOR in the CMIP6 ozone database as a precise representation of the true SOR, but it is likely an improvement compared to the representation in the CMIP5 ozone database.

### 3.4 Atmospheric impact of change in SOR between CMIP5 and CMIP6 ozone databases

We now explore the atmospheric impacts of the differences between the SOR in the CMIP5 and CMIP6 ozone databases using the ECHAM6.3 model sensitivity experiments described in Section 2.3. Figure 9 shows the tropical average annual mean temperature differences in the four perturbation simulations representing 11 year solar cycle maximum conditions with respect to the control simulation representing solar minimum. Note that the tropospheric temperature responses in all sim-

ulations are small because the model includes fixed SSTs and therefore the troposphere does not fully adjust to the imposed solar forcing (e.g. Misios et al. (2016)).

The experiments performed to capture the total (i.e. SSI + SOR) solar cycle impact (dashed lines) show considerable differences in the tropical mean stratospheric temperature response between the recommended CMIP5 (red line) and CMIP6 (blue line) solar forcings. In the CMIP5 case, the maxi-
mum temperature response is around 1.25 K near the stratopause, which can be compared to a much smaller response to the CMIP6 solar forcing inputs of 0.8 K. The SOR-only sensitivity experiments (solid lines) reveal that much of the difference in the total temperature response can be attributed to the differences in the SOR between the CMIP5 and CMIP6 ozone databases. The SOR in the Extended CMIP5 ozone database induces a peak tropical temperature response of 0.85 K (solid red),
which is nearly three times larger than the maximum response to the SOR in the CMIP6 ozone database with an amplitude of 0.3 K (solid blue). In addition to the marked differences in the maximum temperature response, there are also distinct differences in vertical structure. In the CMIP5 case, there is a stronger vertical gradient in the temperature response to the imposed solar forcing, which can be attributed to the highly peaked structure of the SOR in the CMIP5 database at the
stratopause compared to the smoother vertical structure of the SOR in the CMIP6 ozone database (cf. Figures 5(c) and 5(d)). The simulation forced with the SOR from the CMIP6 ozone database also shows a small secondary peak in tropical lower stratospheric temperature of ∼0.3 K due to the presence of a locally enhanced SOR of ∼3%, which is not present in the CMIP5 ozone database. The results show that the change in the representation of the SOR between the recommended CMIP5 and
CMIP6 ozone databases induces a much larger difference in the temperature response between solar cycle minimum and maximum than do changes in the recommended SSI forcing (see also Figure 8 in Matthes et al. (2017)).

The ECHAM6.3 model results help to elucidate the findings of Mitchell et al. (2015), which show a clear difference in the annual mean stratospheric temperature response to the solar cycle between
CMIP5 models that used the CMIP5 ozone database (HadGEM2-CC, MPI-ESM, CMCC) and those with interactive chemistry that simulated their own internally-consistent SOR (CESM1(WACCM), GFDL-CM3, GISS-E2-H, MIROC-ESM-CHEM, MRI-ESM1). Specifically, models that used the CMIP5 ozone database exhibit a markedly larger temperature response near the tropical stratopause, with a stronger vertical gradient, compared to the models with interactive chemistry (see Figure
5 in Mitchell et al. (2015)). One might therefore anticipate that the difference in the stratospheric temperature response between solar cycle minimum and maximum for models with and without interactive chemistry will be smaller in CMIP6 than was found in CMIP5 owing to the fact that the SOR in the CMIP6 ozone database is derived from CCM simulations, albeit without full consistency with the other CMIP6 external forcings such as SSI.

## 4 Conclusions

Changes in stratospheric ozone concentrations are a major part of the atmospheric response to variations in incoming solar radiation over the 11 year solar cycle (e.g. Haigh (1994); Shibata and Kodera (2005); Gray et al. (2009)). The associated solar-ozone response (SOR) must therefore be included in global model simulations to realistically capture the effects of solar variability on the atmosphere.

This study has used a multiple linear regression (MLR) model to analyse the SOR in current satellite observations (Part I; Maycock et al. (2016)) and in global models (Part II). In the present Part II, the SOR is analysed in eight chemistry-climate models (CCMs) from the CCMI-1 project: CCSRNIES-MIROC3.2, CESM1(WACCM), CMAM, CNRM-CM5-3, EMAC(L90), LMDz-REPROBUS-CM5, MRI-ESM1r1, and SOCOL3. These analyses complement earlier studies assessing the SOR in previous generations of CCMs (e.g. Austin et al. (2008); SPARC CCMVal (2010)). In a novel step, we also analyse and compare the SORs in three pre-calculated ozone databases that are prescribed in climate models without interactive chemistry: the Bodeker et al. (2013) Tier 1.4 ozone database and the CMIP5 ozone database (Cionni et al., 2011), which are both based on regression models fit to ozone measurements, and the CMIP6 ozone database, which is created from simulations from two CCMs (CESM1(WACCM) and CMAM).

The CCMI-1 models simulate an annual mean SOR with a peak amplitude of 1-2% in the upper stratosphere ($\sim$3-5 hPa). This is more than a factor of two smaller than the SOR found in SAGE II v6.2 mixing ratio data and is more consistent with results from SAGE II v7.0 and SBUV satellite datasets (Maycock et al., 2016; Dhomse et al., 2016) and with previous CCM studies (e.g. Austin et al. (2008); Sekiyama et al. (2006); Lee and Smith (2003); Egorova et al. (2014); Dhomse et al. (2011, 2016); Hood et al. (2015); SPARC CCMVal (2010)). Many of the CCMI models show larger fractional monthly SORs in the high latitudes during winter and spring, which are likely to be strongly coupled to dynamical processes such as the formation and evolution of the polar vortex. The spread in the best estimate SOR across the CCMI-1 models is around 4 times larger in the tropical lower stratosphere than in the middle and upper stratosphere, and the statistical uncertainties in the SOR are also substantially larger in the lower stratosphere.

There are strong differences in the representations of the SOR in the pre-calculated ozone databases. The peak amplitude of the SOR in the tropics in the CMIP5 and Bodeker ozone databases is substantially larger (5%) than in the CMIP6 database (1.5%). This is because the former databases are derived from observations that include SAGE II v6.2 mixing ratios, which exhibit a larger SOR than found in other satellite ozone datasets (see Part I). In contrast, the CMIP6 ozone database was constructed from CCM simulations and thus its SOR generally resembles that in the CCMI-1 models, both in terms of its broad structure and magnitude and the fact that it exhibits some variation over the annual cycle. Note that the amplitude of the SOR in the CMIP6 ozone database may have been slightly larger if both of the constituent CCMs had used the CMIP6 SSI forcing rather than the NRLSSI-1 forcing from CCMI-1 (Matthes et al., 2017). The CMIP5 database exhibits spurious sharp

horizontal gradients in the SOR across the extratropics, which were alleviated by a simple poleward extrapolation in the Extended CMIP5 ozone database, albeit with considerable uncertainties in the detailed spatial structure. Furthermore, the CMIP5 and Extended CMIP5 ozone databases include a
fixed SOR throughout the year, which is unrealistic.

Sensitivity experiments were performed using a comprehensive global atmospheric model without chemistry (ECHAM6.3) to test how the changes in the recommended SOR and SSI between CMIP5 and CMIP6 affect the simulated annual mean temperature response over the 11 year solar cycle. The experiments show that changes in the SOR between CMIP5 and CMIP6 cause a decrease in
the tropical average temperature response over the solar cycle of up to 0.6 K, or around 50% of the total amplitude. This impact on the simulated stratospheric temperature response over the solar cycle is many times larger than the separate impact (i.e. without ozone feedbacks) of changes in the recommended SSI forcing between CMIP5 and CMIP6. The results indicate that differences in the representation of the SOR amongst CMIP5 models is likely to be a major explanatory factor for the
large spread in the stratospheric temperature responses to the solar cycle found in CMIP5 models (Mitchell et al., 2015). The broader relevance of different representations of the SOR for atmospheric dynamics and regional surface climate responses to the solar cycle remains to be explored. However, Hood et al. (2015) suggested CMIP5 models with an interactive representation of the SOR showed a stronger high latitude dynamical response to the solar cycle.

Parts I and II of this study have shown that uncertainties remain in understanding the SOR, which present a challenge for including these effects in model simulations. However, given the inclusion of variations in the SOR over the annual cycle, as well as the greater consistency of the amplitude of the SOR with CCM results, CMIP6 models without chemistry are encouraged to use the recommended CMIP6 ozone database. The CMIP6 solar-ozone coefficients are published with this paper
(https://doi.org/10.5518/348), and have already been used in other modelling projects such as PMIP4 (Jungclaus et al., 2017). Nevertheless, whatever approach is employed, all CMIP6 modelling groups are encouraged to document the representation of the SOR and SSI in their simulations to facilitate future analysis of solar-climate impacts.

*Acknowledgements.*  ACM acknowledges funding from an AXA Postdoctoral Fellowship, the ERC ACCI Grant
Project No. 267760, and a NERC Independent Research Fellowship (NE/M018199/1). ACM also acknowledges funding from the COST action ES1005 Towards a more complete assessment of the impact of solar variability on the Earth's climate (TOSCA) for a Short-term Scientific Mission to GEOMAR in September 2014 which initiated this work. We are grateful to support for scientific meetings from the WCRP/SPARC SOLARIS-HEPPA Activity. Parts of the work at GEOMAR Helmholtz Centre for Ocean Research Kiel were performed within the
Helmholtz-University Young Investigators Group NATHAN, funded by the Helmholtz-Association and GEO-MAR. H.A. acknowledges Environment Research and Technology Development Fund of the Environmental Restoration and Conservation Agency, Japan (2-1709) and NECSX9/A(ECO) computers at CGER, NIES. The National Center for Atmospheric Research (NCAR) is sponsored by the U.S. National Science Foundation

(NSF). WACCM is a component of the Community Earth System Model (CESM), which is supported by NSF and the Office of Science of the U.S. Department of Energy. The EMAC simulations have been performed at the German Climate Computing Centre (DKRZ) through support from the Bundesministerium für Bildung und Forschung (BMBF). DKRZ and its scientific steering committee are gratefully acknowledged for providing the HPC and data archiving resources for the consortial project ESCiMo (Earth System Chemistry integrated Modelling). The SOCOL team acknowledges support from the Swiss National Science Foundation under grant agreement CRSII2_147659 (FUPSOL II). RT acknowledges his funding by the LABEX L-IPSL project (grant ANR-10-LABX-18-01). SB has been partially supported by the European project StratoClim (603557 under programme FP7-ENV.2013.6.1-2). We acknowledge the modelling groups for making their simulations available for this analysis, the joint WCRP SPARC/IGAC Chemistry-Climate Model Initiative (CCMI) for organizing and coordinating the model data analysis activity, and the British Atmospheric Data Centre (BADC) for collecting and archiving the CCMI model output. We are grateful to Greg Bodeker (Bodeker Scientific) and Birgit Hassler (NOAA) for providing the combined vertical ozone profile database from Bodeker et al. (2013).

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

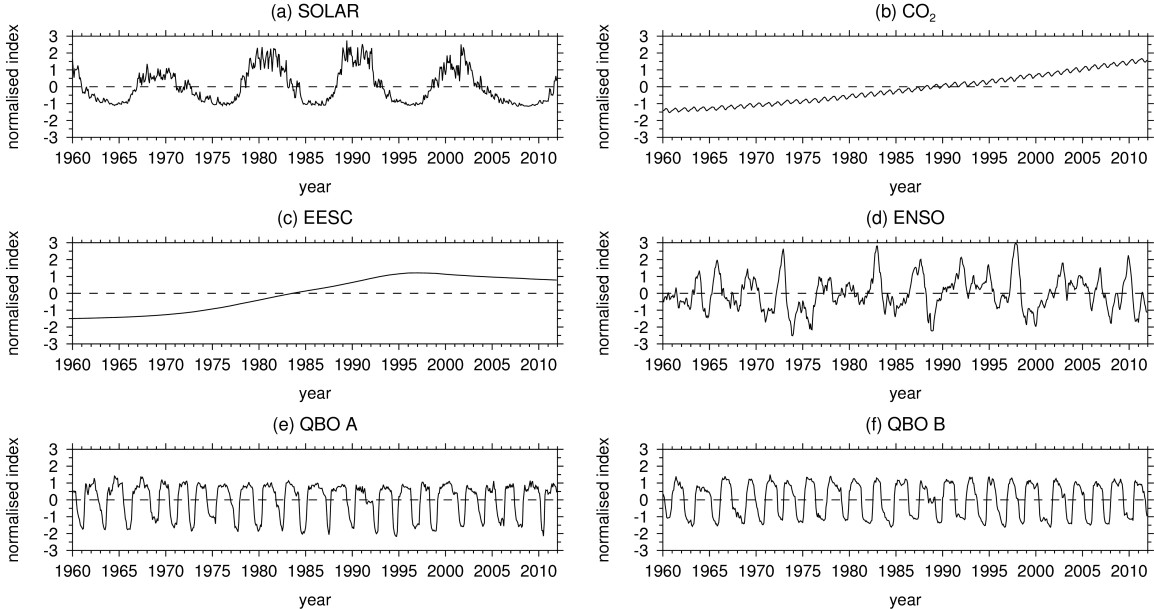

**Figure 1.** Timeseries of the six basis functions used in the MLR analysis. (a) Solar forcing based on F10.7cm flux; (b) $CO_2$; (c) equivalent effective stratospheric chlorine; (d) ENSO index; (e, f) two orthogonal QBO indices defined as the first two principal component timeseries of tropical zonal mean zonal winds (in this case taken from observations). The timeseries are in units of standard deviation.

Yukimoto, S., Yoshimura, H., Hosaka, M., Sakami, T., Tsujino, H., Hirabara, M., Tanaka, T. Y., Deushi, M., Obata, A., Nakano, H., Adachi, Y., Shindo, E., Yabu, S., Ose, T., and Kitoh, A. Meteorological Research Institute Earth System Model Version 1 (MRI-ESM1) – Model Description. *Tech. Rep. of MRI*, 64:83pp., 2011.

Yukimoto, S., Adachi, Y., Hosaka, M., Sakami, T., Yoshimura, H., Hirabara, M., Tanaka, T. Y., Shindo, E., Tsujino, H., Deushi, M., Mizuta, R., Yabu, S., Obata, A., Nakano, H., Koshiro, T., Ose, T., and Kitoh, A. A new global climate model of the Meteorological Research Institute: MRI-CGCM3 – Model description and basic performance. *J. Meteorol. Soc. Jpn.*, 90:23-64, 2012.

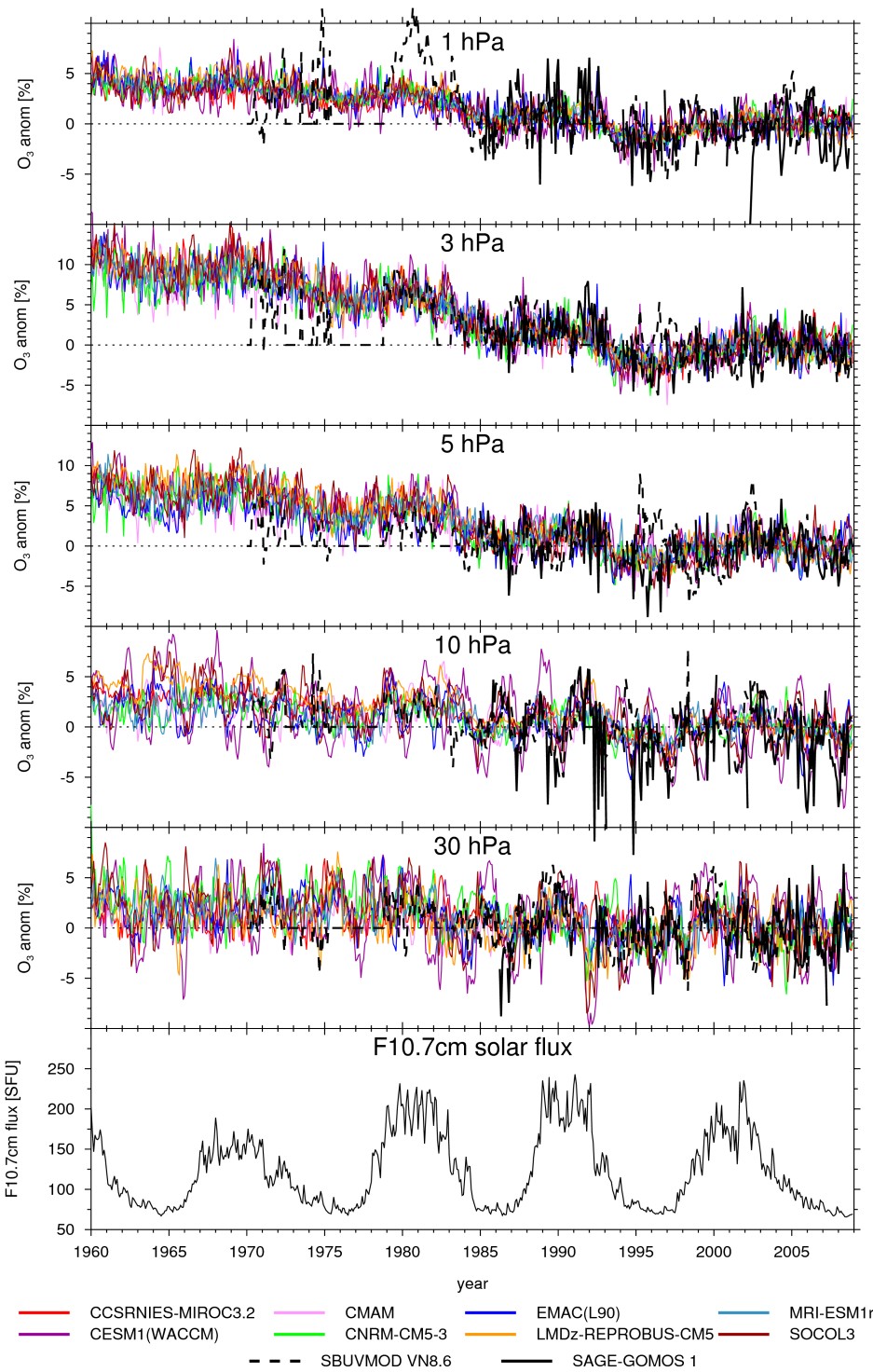

**Figure 2.** Timeseries of deseasonalised percent tropical (30°S-30°N) ozone anomalies in CCMI-1 models for 1960-2009 and two satellite datasets at 1 hPa, 3 hPa, 5 hPa, 10 hPa and 30 hPa. The lowest panel shows the F10.7 cm solar flux for reference. Anomalies are shown relative to a baseline period 1985-2009.

# 1960−2009 Annual Ozone Response [%]

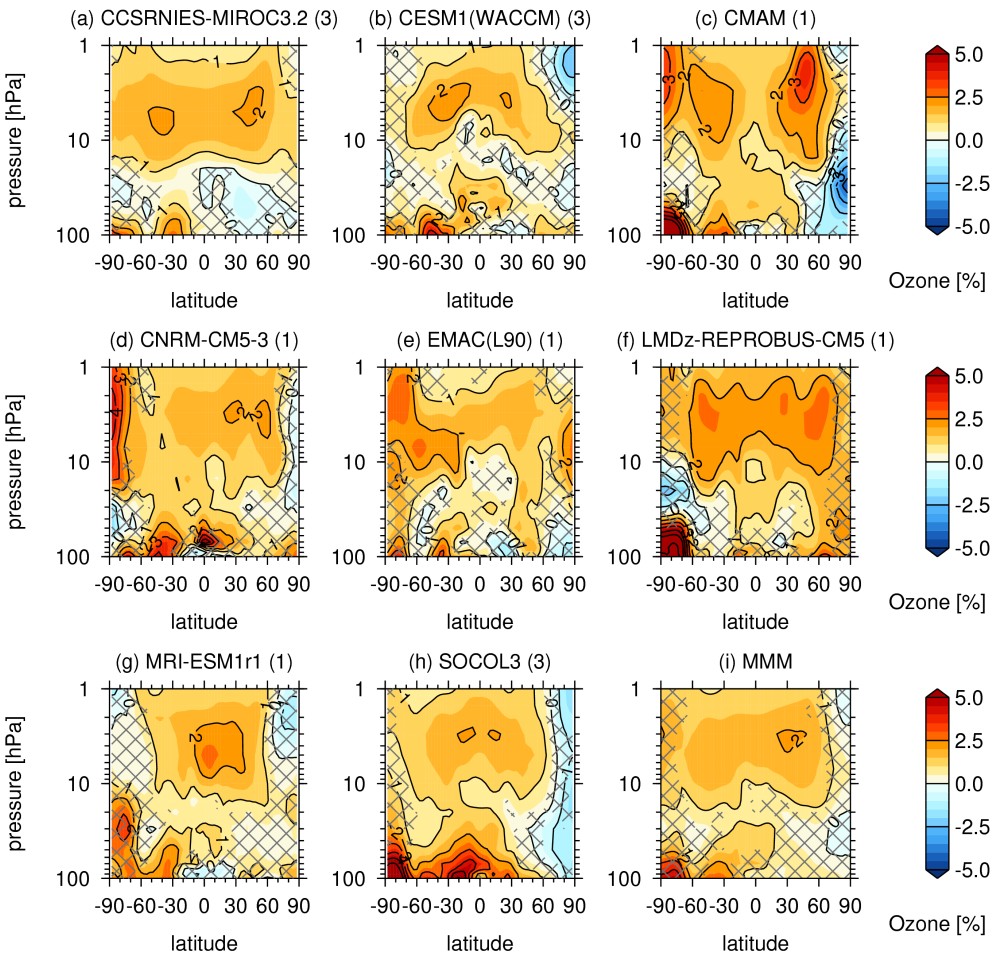

**Figure 3.** The percent (%) differences in stratospheric ozone mixing ratios per 130 SFU for 1960-2009 in the CCMI-1 models listed in Table 1. The solid contours denote 1% intervals. Hatching denotes regions where the regression coefficients are not significantly different from zero at the 95% confidence level. Panel (i) shows the multi-model mean (MMM) with hatching denoting where the MMM response is smaller than $\pm 2$ sd of the intermodel spread.

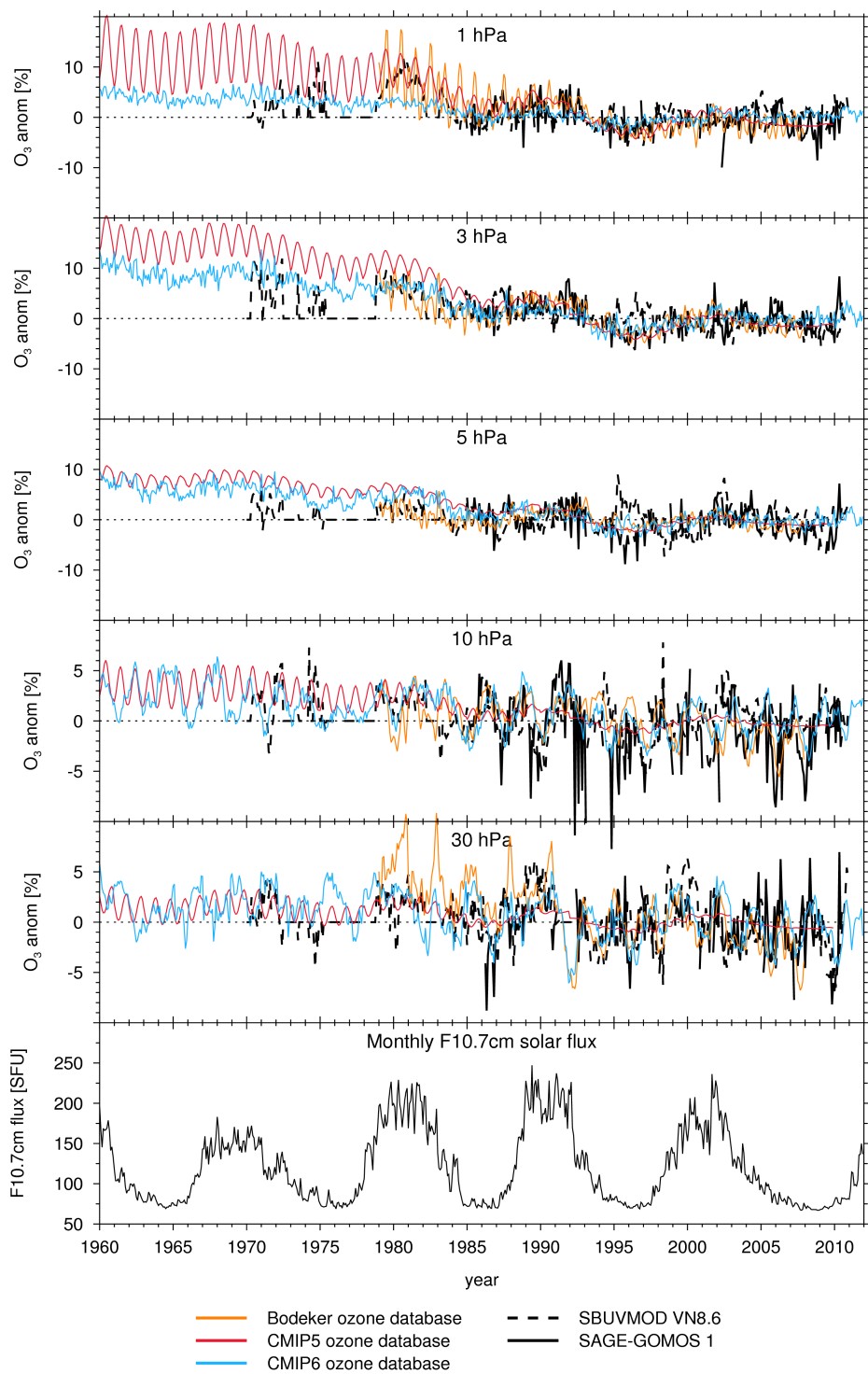

**Figure 4.** Timeseries of deseasonalised percent tropical (30°S-30°N) ozone anomalies from two satellite observation datasets (black) and the Bodeker (orange), CMIP5 (Cionni et al., 2011) (red), and CMIP6 (blue) ozone databases over the period 1960-2011 at (a) 1 hPa, (b) 3 hPa, (c) 5 hPa, (d) 10 hPa and (e) 30 hPa. The lowest panel shows the F10.7 cm solar flux for reference. Anomalies are shown relative to a baseline period 1985-2009.

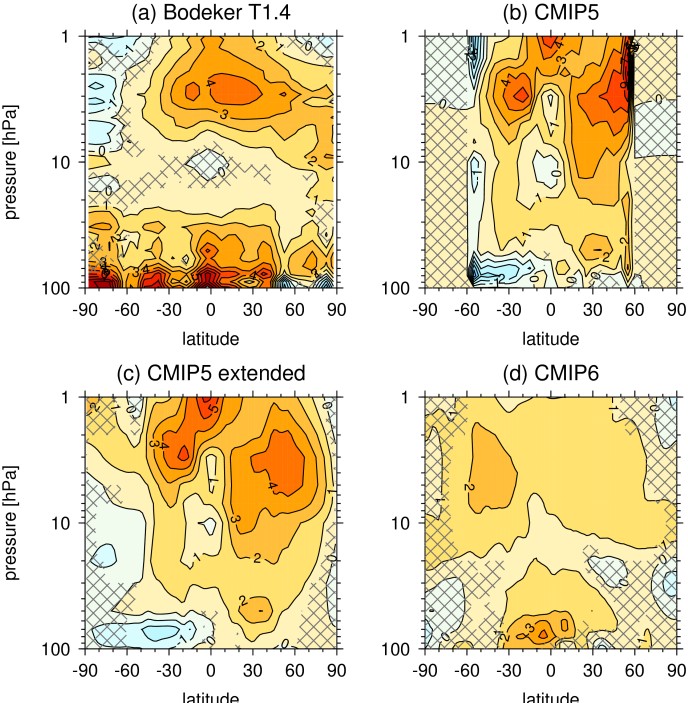

**Figure 5.** The annual mean percent (%) differences in ozone per 130 SFU for the (a) Bodeker (1979-2007), (b) CMIP5 (1960-2005), (c) Extended CMIP5 (1960-2005) and (d) CMIP6 (1960-2011) ozone databases. The contour interval is 1%. The hatching in (d) is as in Figure 3.

# Tropical mean SOR [%]

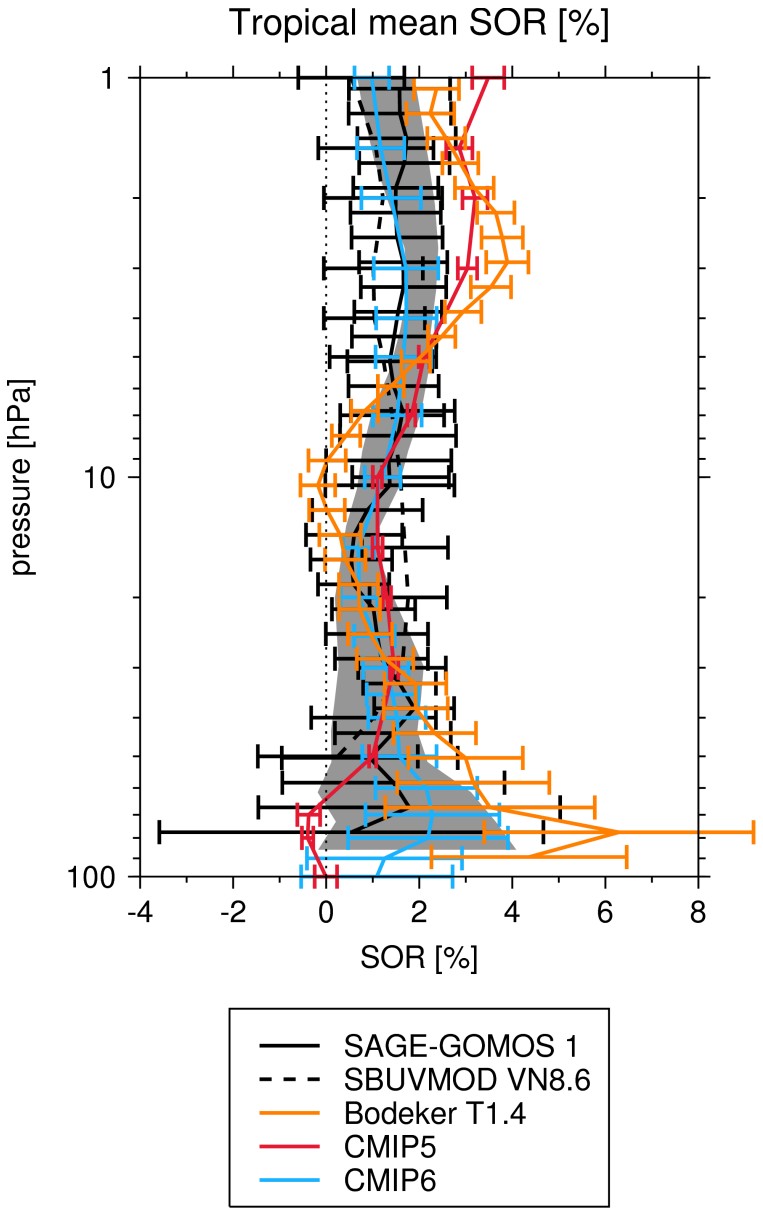

**Figure 6.** Vertical profiles of the tropical (30°S-30°N) average annual SOR per 130 SFU (%). The range of the best estimates across the eight CCMI-1 models is shown in the dark grey shading. The lines show the tropical mean annual SOR in the three climate model ozone databases discussed in Section 3.2 and two satellite ozone datasets from Maycock et al. (2016) (SBUVMOD VN8.6 and SAGE-GOMOS 1). The whiskers denote 2.5-97.5% confidence intervals on the estimated SOR.

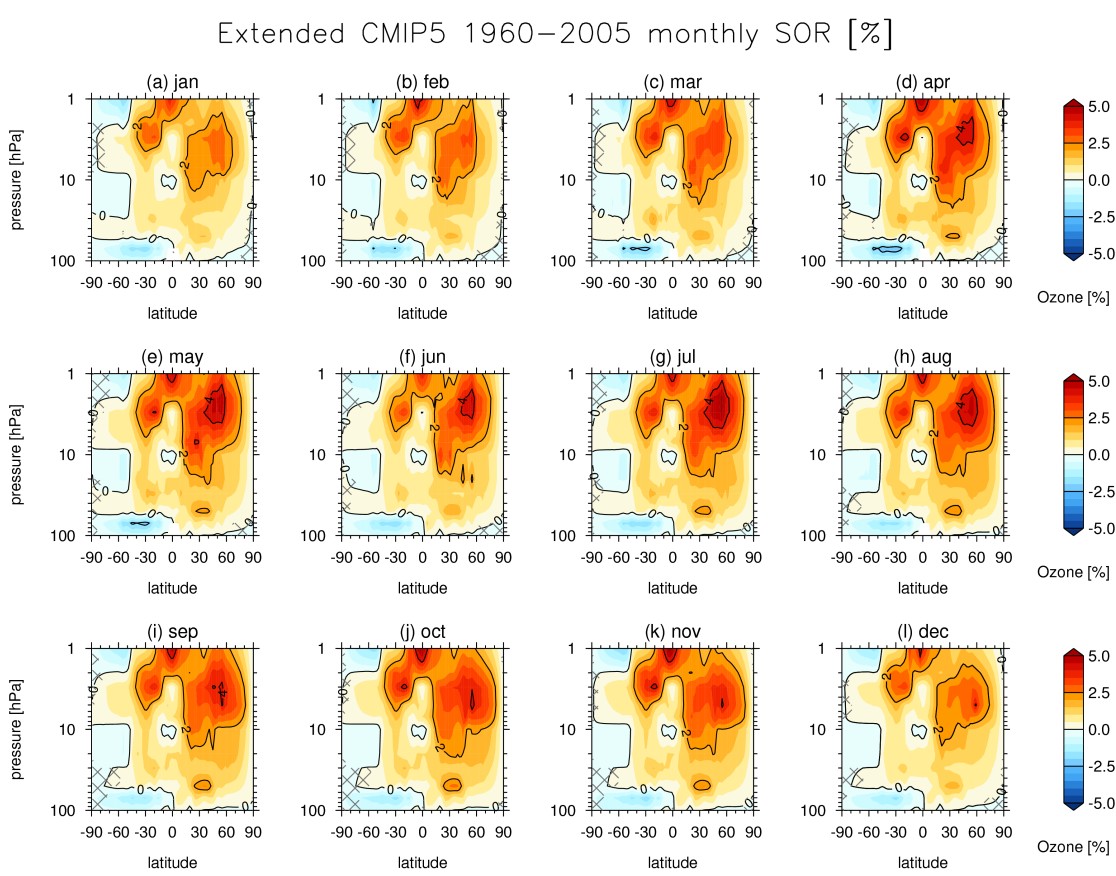

**Figure 7.** Monthly mean percent (%) ozone anomalies per 130 SFU for (a) January to (l) December in the Extended CMIP5 ozone database. The solid contours denote 2% intervals.

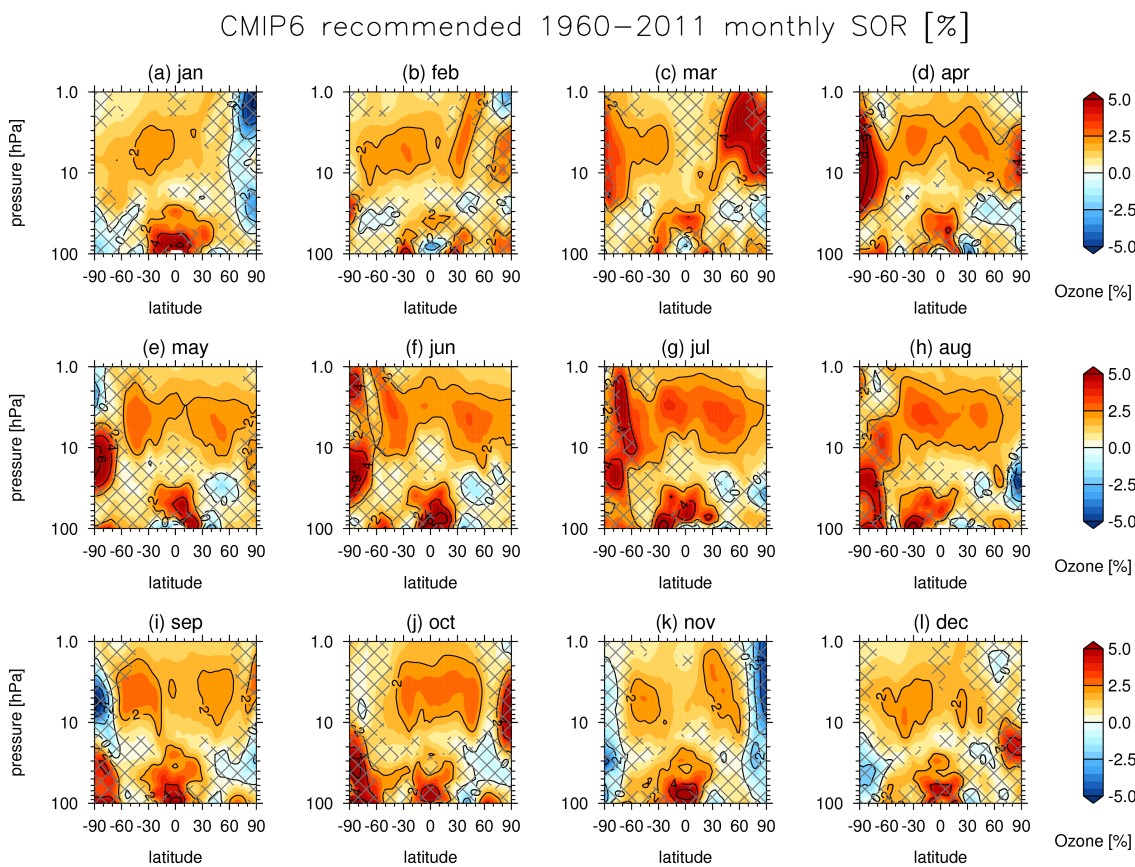

**Figure 8.** Monthly mean percent (%) ozone anomalies per 130 SFU for (a) January to (l) December in the CMIP6 ozone database. The solid contours denote 2% intervals. Hatching denotes regions where the regression coefficients are not significantly different from zero at the 95% confidence level.

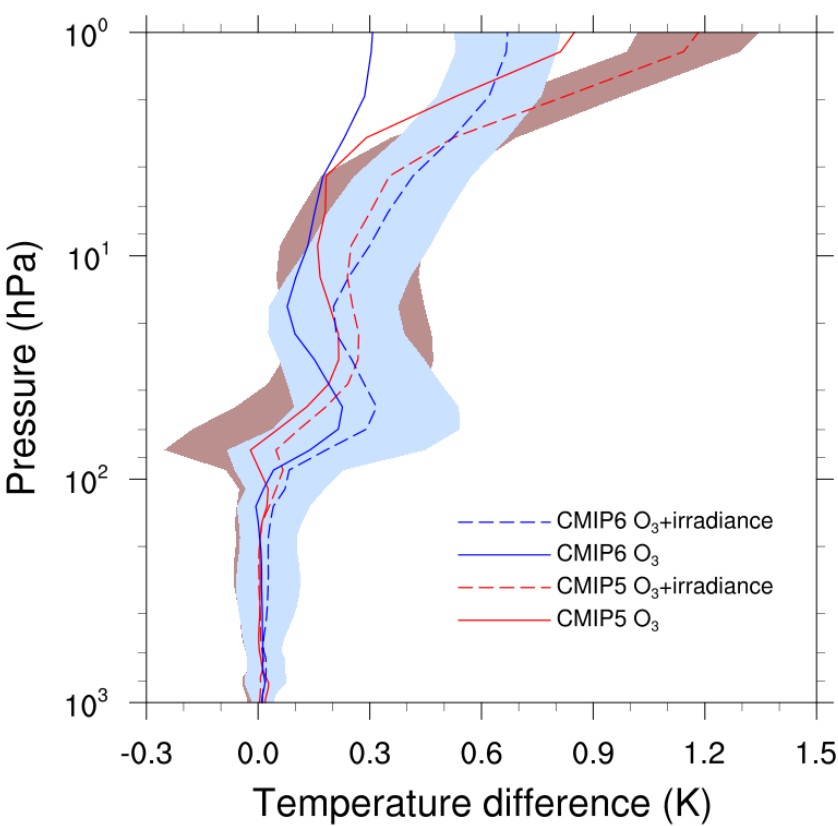

**Figure 9.** Average tropical (30°S-30°N) solar cycle (max-min) temperature anomalies as simulated by ECHAM6. Anomalies have been calculated between four sensitivity experiments representing different solar maximum conditions and a reference experiment representing solar minimum conditions. The sensitivity experiments are performed by prescribing: (red solid) SOR from the Extended CMIP5 ozone database; (red dashed) recommended SOR and solar spectral irradiance anomalies for CMIP5; (blue solid) historical SOR from recommended CMIP6 ozone database; and (blue dashed) recommended SOR and solar spectral irradiance anomalies for CMIP6. The shaded regions denote 2.5-97.5% confidence intervals on the combined forcing responses.

| Model | No. ensembles | QBO | No. shortwave bands | Reference |
|---|---|---|---|---|
| CMAM | 3 | No | 4 | Jonsson et al. (2004); Scinocca et al. (2008) |
| CESM1(WACCM) | 3 | Nudged | 19 | Marsh et al. (2013); Solomon et al. (2015) |
| CCSRNIES-MIROC3.2 | 3 | Nudged | 20 | Imai et al. (2011); Akiyoshi et al. (2016) |
| CNRM-CM5-3 | 1 | No | 6 | Voldoire et al. (2011); Michou et al. (2011); http://www.cnrm-game-meteo.fr/ |
| EMAC(L90) | 1 | Nudged | 55 in the stratosphere (<70 hPa) | Jöckel et al. (2016) |
| LMDz-REPROBUS-CM5 (L39) | 1 | No | 2 | Marchand et al. (2011); Szopa et al. (2013); Dufresne et al. (2013) |
| MRI-ESM1r1 | 1 | Internal | 22 | Yukimoto et al. (2011, 2012); Deushi and Shibata (2011) |
| SOCOL3 | 3 | Nudged | 6 | Stenke et al. (2013); Revell et al. (2015) |

**Table 1.** Details of the CCMI-1 models used in this study and the number of ensemble members available for the REFC1 experiment for the period 1960-2009. See Morgenstern et al. (2017) for more details.