# Peer review of "The representation of solar cycle signals in stratospheric ozone. Part II: Analysis of global models"

_Atmospheric Chemistry and Physics, 2017_

## Short Comment (SC1) · 31 May 2017

Dear authors,

Thank you for using IGAC/SPARC CCMI model data for your research.

As stated in the CCMI data policy, could you please add the following statement to your acknowledgment?

*We acknowledge the modeling groups for making their simulations available for this analysis, the joint WCRP SPARC/IGAC Chemistry-Climate Model Initiative (CCMI) for organizing and coordinating the model data analysis activity, and the British Atmospheric Data Centre (BADC) for collecting and archiving the CCMI model output.*

[Figure]

Likewise CCMI data policy, the following reference to the CCMI archive is missing:

*Hegglin, M. I. and J.-F. Lamarque (2015); The IGAC/SPARC Chemistry-Climate Model Initiative Phase-1 (CCMI-1) model data output, NCAS British Atmospheric Data Centre, date of citation. http://catalogue.ceda.ac.uk/uuid/9cc6b94df0f4469d8066d69b5df879d5*

---

## Short Comment (SC2) · 5 Jun 2017

Thank you for these comments, we will include the additional citations in the revised manuscript.

---

## Referee Comment (RC1) · JD Haigh (Referee) · 13 Jun 2017

The paper presents an analysis of the responses found in the ozone fields of coupled chemistry-climate models to specification of solar spectral irradiance, how these compare to the ozone fields prescribed in IPCC climate models (and to the signals found in observational datasets). The work is carefully planned and thorough and provides a suitable background against which future work can be planned and studied.

l.44 Tell reader that Maycock et al. (2016) is Part I

l.121 Why only one ensemble member? The modellers have done several to provide

you with stats!

l.179 ref l.671 Matthes (2017) now published (though with a weak explanation for the political expediency involved in the averaging of two datasets!).

l.184 Clarify "up∼0.3%": presumably not 0.3% of signal but 0.3% on top of c. 2% (?)

l.233 Have you looked at the impact of this choice of AR model (cf none or AR(1))?

l.300, l.330 and elsewhere. Comparison to observational results of Part I interesting but difficult to extract from this text.

l.349 et seq. Useful comment on effect of QBO simulations.

l.391-2 Indeed! Can you make any judgement on what is causing these differences between datasets?

l.425 Good point re seasonality.

l.491-494. Any conclusion on why these models produce a signal in the tropical lower stratosphere?

l.515 Not sure that you have justified the statement that proper SOR and SSI are needed for solar-climate impacts. Of course I agree with that (!) but you have not discussed climate (troposphere) much at all in this paper, and only used models with fixed SSTs.

l.533 A reasonable summary paragraph but it is a bit disappointing that we seem to be no nearer any understanding of the solar signal and the conclusion is just that more data is needed.

---

## Referee Comment (RC2) · Anonymous Referee #2 · 23 Jun 2017

Overview of study

The authors look at how the ozone (mainly stratospheric) changes in response to solar cycle activity (not including many secondly solar effects, such as high energy particles). The use CMIP5 and 6 data, and compare with observations, mainly characterised in part 1 of these papers.

I find the study comprehensive in its analysis, but not particularly novel in terms of the science, and certain not novel in terms of increased scientific understanding. The study essentially regresses out the solar signal from ozone in climate models, which has been done to death. I appreciate a lot of work has gone in to applying it to a new

data set, but I can see little advancement in scientific knowledge in what is done. The authors final summary seems testament to this, where their conclusion is essentially 'we need more data'. The scientific analysis is far from rigorous as well, with statistical significance very rarely performed, in unclear for the figures where it has been done.

Concerns (major)

- The novelty to the study. Many studies have done very similar things to this study. Some of these studies are cited in the main text, but it is often not clear to a reader who is unfamiliar with the literature just how similar these studies really are. In many cases reproducing very similar figures. The authors need to be clear up front what is new here, and attribute all the repeated information to the correct papers.

- The statistical significance in this study. This is very poor, and often non-existent. The authors need to be clear about what their significance test is, what it is showing, and most importantly, they actually need to do some significance testing for most of the plots. In some plots, different significance tests will be needed for each panel. i.e. Figure 3, a different test will be needed for the individual model, as for the MMM. I am not even sure if the MMM has significance in the current study?

- Regression methodology. The methodology the authors use has no measure of uncertainty in the basis functions. This is a fundamental problem, because some of the basis functions have a good deal of uncertainty associated with them. The authors should add this uncertainty in to better reflect the uncertainty in the final result. The regression method they use is cited as from Maycock et al, but in reality, it probably has roots in far earlier solar-regression studies (see my first point, of giving due where it is deserved, even if methods/results are slightly different). I expect the authors arguments to not including uncertainty in the basis functions will be 'it has been done multiple times before', and cite a number of studies. However, this does not mean it is correct, unfortunately. A feel at some point, one of these studies need to include these basis uncertainties, at the very least to show that it doesn't make a difference (although

I expect that it does).

- Anomalies. Often the authors use anomalies of variables, rather than the absolute variables. It would be good to see who real values of the data. I realise this can not always be done, but in some figures, for instance Figure 4, this would be very informative. Anomalies often make things look better!

- There is a lot of focus on the CMIP6 ozone data set. But seems to be absolutely no citation to documentation on this data set. I note that the creators of the data set are not authors on this paper, and perhaps some of that lack of knowledge is reflected in the text. Is there a CMIP6 ozone paper coming out? Should this current paper be kept out of publication till that exists? This should certainly be true if there is any overlap.

Concerns (minor)

1. Line 1: This alone would not fully capture the response

2. The SOR seems a little strange in this context, because it is not obvious (until later) that the SOR is not a 'set thing', it is only known within uncertainty bounds (and so different CCM give different SORs).

3. Line 9: . . . ozone databases' – at this point it is not clear if the ozone data basis is the prescribed ozone, or simulated ozone from a CCMI.

4. Line 11 Make clear that you refer to historical period ozone

5. Line 13: weak compared with what?

6. Line 76 – a citation is really needed here (see major concern).

7. Line 89 – what time frequency is this data?

8. Line 94 'transferring' – please revise this word.

9. L124. Why only 1 ensemble member? Please repeat with all of them. You need to capture the uncertainty.

10. Page 5 (top): This seems very similar to Hood et al, please state that.

11. Line 37: 5 x 5degree. This is not normal, why has the interpolation taken place?

12. Equ 1: Please cite where this came from originally.

13. Equ 1: Do the authors have any views on the breakdown between the long terms solar response, and the 11-year response?

14. Figure 1: Surely these QBO signals are just from one model? These will change.

15. Line 218-219: 'better proxy' not convincing to me. Please cite a paper that compares these.

16. Line 220-225: This section on the volcanic signal is vague. The data sets the authors use are not long, in fact some figure just use ~30 years of data. Very short for regression. Volcanic signals will cause issues in the regression, and it is not clear the authors have dealt with the properly (nor in Part 1).

17. Line 240. I think the authors need to show the autocorrelation plots to the reviewers, so we can assess this evidence. I agree they probably do not need to go in the main text.

18. Section 2.3: please describe more how this model fits into the wider models of CMIP5. Then we can assess suitability.

19. Line 255-260: This is worrying that the lower signal might not be so well captured.

20. Figure 2: Please just plot the SAGE2 and SBUV observations on this plot.

21. Line 291-295: power spectra would be useful here.

22. Line 350-358: This is an important point the authors make. Are you saying this is a drawback of the CMIP6 ozone data set? Please expand on your recommendations here.

23. Figure 3: What is the significance test here? Does the MMM have significance?

24. Figure 3: Why are tropospheric values masked out?

25. Figure 4. Colors very similar

26. Figure 5. Is there any significance on here? At this point (analysis of figure 5-9), I do not believe it constructive to have an in-depth review, because the significance is mainly missing, or hard to understand. You are interpreting potentially small signals compared to the noise.

27. Line 438-440: I think this is wrong, the SSTs do not constrain the upper tropospheric temperatures this much!

---

## Author Comment (AC1) · 4 Dec 2017

**Author response to review by Joanna D. Haigh**

The representation of solar cycle signals in stratospheric ozone. Part II: Analysis of global models" by Amanda C. Maycock et al.

The paper presents an analysis of the responses found in the ozone fields of coupled chemistry-climate models to specification of solar spectral irradiance, how these compare to the ozone fields prescribed in IPCC climate models (and to the signals found in observational datasets). The work is carefully planned and thorough and provides a suitable background against which future work can be planned and studied.

We thank the reviewer for her supportive comments on the manuscript. We reply to her specific points below.

**_Minor comments_**

l.44 Tell reader that Maycock et al. (2016) is Part I
Added

l.121 Why only one ensemble member? The modellers have done several to provide you with stats!
We have updated the analysis to use all available ensemble members for the models.

l.179 ref l.671 Matthes (2017) now published (though with a weak explanation for the political expediency involved in the averaging of two datasets!).
Reference updated

l.184 Clarify "up_0.3%": presumably not 0.3% of signal but 0.3% on top of c. 2% (?)
Text has been changed for clarification

l.233 Have you looked at the impact of this choice of AR model (cf none or AR(1))?
Figure R1 below shows the decorrelation timescale for the MLR model residuals. The e-folding time is >2 months in some regions of the middle and lower stratosphere. Figure R2 below is as in Figure 3 of the main text but assuming no AR model for the residuals. The results are similar to those using an AR(2) model, with the main exception found in the polar lowermost stratosphere.  Therefore to avoid giving potentially misleading information about the SOR in the polar lowermost stratosphere we have restricted the plots in the revised manuscript to a maximum pressure of 100 hPa.

[Figure]

Figure R1: e-folding time [in months] of the ACF in the MLR residuals for the CCMI-1 models.

1960−2009 Annual Ozone Response [%]

[Figure]

Figure R2: As in Figure 3 of the main text but assuming no AR model.

l.300, l.330 and elsewhere. Comparison to observational results of Part I interesting but difficult to extract from this text.

We have added timeseries of ozone anomalies from two satellite observation datasets described in Part I to Figures 2 and 4 to facilitate the comparison with results from Part I.

l.391-2 Indeed! Can you make any judgement on what is causing these differences between datasets?
l.491-494. Any conclusion on why these models produce a signal in the tropical lower stratosphere?

Both comments relate to the presence or absence of a significant SOR in the tropical lower stratosphere. There are some methodological sensitivities to the robustness of this feature. For example, this is one of the main regions where sensitivity to the choice of AR model is seen. This is particularly found in models where the regression residuals show long autocorrelation timescales in the tropical lower stratosphere (see e.g. SOCOL3 in Figure R1 above and compare Figure R2(h) and Figure 3(h) from the main text). Consequently, the estimated uncertainties in the magnitude of the SOR in the tropical lower stratosphere are larger than in the upper stratosphere (see e.g. Figure 6). In the revised manuscript, we have also altered the approach for accounting for volcanic effects by excluding 2 year periods following eruptions rather than including a volcanic term in the MLR. This also has a modest effect on the SOR in the tropical lower stratosphere in some models (compare Figure 3 in revised manuscript with original Figure 3).

Aside from the above methodological issues, additional analysis (not shown) has been performed on the Transformed Eulerian Mean residual vertical velocity fields for the models that provide this data for the refC1 experiment (CCSRNIES-MIROC3.2, CMAM, EMAC, MRI-ESMr1, SOCOL3). There is no evidence of a significant weakening in tropical lower stratospheric upwelling in the models that show some enhancement in the SOR in this region, as has been suggested in some earlier studies.

The following text has been added to the manuscript:
"One of the CCMI-1 models (SOCOL3) appears to show an enhanced SOR in the tropical lower stratosphere, which is similar in amplitude to that seen in some CCMVal-1 models. However, this feature shows some sensitivity to the choice of autoregressive model in the MLR model probably because the decorrelation timescale for the regression residuals in the tropical lower stratosphere is longer than two months in SOCOL3 and some of the other CCMs (not shown). Further analysis of the Transformed Eulerian Mean residual vertical velocity does not reveal a substantial change in the rate of upwelling in the tropical lower stratosphere in any of the models (not shown)."

l.515 Not sure that you have justified the statement that proper SOR and SSI are needed for solar-climate impacts. Of course I agree with that (!) but you have not discussed climate (troposphere) much at all in this paper, and only used models with fixed SSTs.
This sentence does not claim that proper SOR and SSI are needed to simulate solar-climate impacts, rather it is simply a request to CMIP6 modellers to document the implementation of SSI and the SOR to enable interpretation of the model output after the experiments are finished. For example, a model that does not include any representation of the SOR might be expected to have a weaker atmospheric response to the solar cycle than a model that does include a SOR. CMIP models are often set up in different ways and traceability can be a challenge. This statement in the manuscript is

therefore only intended as an appeal for documentation on how CMIP6 models implement the SOR and SSI in order to interpret model differences once data become available. We have therefore left the text as before.

l.533 A reasonable summary paragraph but it is a bit disappointing that we seem to be no nearer any understanding of the solar signal and the conclusion is just that more data is needed.
The last part of the conclusions has been edited to focus on the new findings of the study. The last paragraph now reads: "Parts I and II of this study have shown that uncertainties remain in understanding the SOR, which present a challenge for including these effects in model simulations. However, given the inclusion of variations in the SOR over the annual cycle, as well as the greater consistency of the amplitude of the SOR with CCM results, CMIP6 models without chemistry are encouraged to use the recommended CMIP6 ozone database in order to potentially improve the atmospheric response to the solar signal. Nevertheless, whatever approach is employed, all CMIP6 modelling groups are encouraged to document the representation of the SOR and SSI in their simulations to facilitate future analysis of solar-climate impacts."

---

## Author Comment (AC2) · 4 Dec 2017

**Author response to review by Anonymous Referee 2**

The representation of solar cycle signals in stratospheric ozone. Part II: Analysis of global models" by Amanda C. Maycock et al.

**Overview of study**

The authors look at how the ozone (mainly stratospheric) changes in response to solar cycle activity (not including many secondly solar effects, such as high energy particles). The use CMIP5 and 6 data, and compare with observations, mainly characterised in part 1 of these papers. I find the study comprehensive in its analysis, but not particularly novel in terms of the science, and certain not novel in terms of increased scientific understanding. The study essentially regresses out the solar signal from ozone in climate models, which has been done to death. I appreciate a lot of work has gone in to applying it to a new data set, but I can see little advancement in scientific knowledge in what is done. The authors' final summary seems testament to this, where their conclusion is essentially 'we need more data'. The scientific analysis is far from rigorous as well, with statistical significance very rarely performed, in unclear for the figures where it has been done.

We thank the reviewer for his/her detailed comments on the manuscript, which we address below. While the reviewer has raised some criticisms, which we have addressed and which have helped to improve the manuscript, we firmly believe that the manuscript contains relevant new results that will be of interest to the broad atmospheric and solar research communities, and therefore that the manuscript warrants publication in ACP.

**Concerns (major)**

1) The novelty to the study. Many studies have done very similar things to this study. Some of these studies are cited in the main text, but it is often not clear to a reader who is unfamiliar with the literature just how similar these studies really are. In many cases reproducing very similar figures. The authors need to be clear up front what is new here, and attribute all the repeated information to the correct papers.

a) Our study compares the representation of the solar-ozone response (SOR) in models with interactive chemistry (CCMs) against the prescribed SOR in GCMs. To our knowledge such a comparison has not been been performed before and therefore comprises an important advance to the field. It is particularly relevant for putting into context recent multi-model studies (e.g. Mitchell et al., 2015; Hood et al., 2015) that include CCMs and/or GCMs. Our results show that the representation of the SOR is crucial (arguably more important than changes to the SSI forcing dataset -- see Matthes et al (2017)) for determining differences in modeled solar cycle responses between CMIP5 and CMIP6. We deem these to be sufficiently interesting and important conclusions to justify publication in ACP.

b) To clarify that multiple regression methods have been widely employed to extract solar cycle variations in ozone datasets before, we have added the following text at the start of Section 2.2:

"Multiple linear regression models have been used to analyse drivers of secular trends and variability in stratospheric ozone for many decades (see e.g. Staehelin et al., 2001 and references therein). In the context of extracting solar cycle variability from ozone timeseries, there is a long history of similar methods being applied to both satellite observations (e.g., Soukharev and Hood, 2006; Remsberg 2008; Tourpali et al 2007; Remsberg and Lingenfelser, 2010; Dhomse et al 2016; Lee and Smith, 2003; Lean 2014; Randel and Wu, 2007; Merkel et al 2011) and chemistry-climate models (Austin et al., 2008; Sekiyama et al., 2006; Lee and Smith, 2003; Egorova et al., 2004; Dhomse et al., 2011; Dhomse et al., 2016; Hood et al., 2015; SPARC CCMVal, 2010). Here we follow the methodology described by Maycock et al (2016), which is very similar to the methods described in those earlier studies."

We have also edited the Introduction to state an explicit set of novel objectives for the study:

"The objectives of this study are therefore:

- to provide an update to previous CCM studies by analysing the SOR in CCMI-1 models.
- to evaluate the SOR in three pre-calculated ozone databases for climate models from CMIP5, CMIP6 and Bodeker et al (2013).
- to compare the CCMs and ozone databases with satellite observations from Part I (Maycock et al, 2016).
- to perform atmospheric model experiments to quantify the impact of differences in the SOR between CMIP5 and CMIP6 on the simulated atmospheric response to the 11 year solar cycle.

Collectively these objectives provide a comprehensive assessment of the represention of the SOR in current CCMs and global climate models."

We have also added at appropriate points in the results section statements connecting our results to figures in earlier multi-model studies such as Austin et al (2008) and Hood et al (2015). While the CCMI-1 model analysis is an update on these earlier studies, the explicit comparison with pre-calculated ozone fields is new.

**We hope that the reviewer agrees these changes adequately acknowledge the earlier work that our study builds on.**

2) The statistical significance in this study. This is very poor, and often non-existent. The authors need to be clear about what their significance test is, what it is showing, and most importantly, they actually need to do some significance testing for most of the plots. In some plots, different significance tests will be needed for each panel. i.e. Figure 3, a different test will be needed for the individual model, as for the MMM. I am not even sure if the MMM has significance in the current study?

The reviewer is correct that in the original manuscript the MMM result in Figure 3i did not show any estimate of statistical significance. This has been added in the revised manuscript based on regions where the MMM response is smaller than ±2 standard deviations of the intermodel spread derived from Figures 3(a-h).

Hatching denoting regions where the central estimate of the regression coefficients is not statistically significant at the 95% confidence level has also been added to Figures 5 and 7.

We have added shading to Figure 9 denoting  $\pm 2$  standard deviations of the interannual variations in temperature over the 50 year experiments as an estimate of the 2.5-97.5% confidence intervals for the ECHAM6.3 modelled responses.

All figures in the revised manuscript (with the exception of the raw timeseries) therefore now include appropriate estimates of the statistical significance of the results.

3) Regression methodology. The methodology the authors use has no measure of uncertainty in the basis functions. This is a fundamental problem, because some of the basis functions have a good deal of uncertainty associated with them. The authors should add this uncertainty in to better reflect the uncertainty in the final result. The regression method they use is cited as from Maycock et al, but in reality, it probably has roots in far earlier solar-regression studies (see my first point, of giving due where it is deserved, even if methods/results are slightly different). I expect the authors arguments to not including uncertainty in the basis functions will be 'it has been done multiple times before', and cite a number of studies. However, this does not mean it is correct, unfortunately. I feel at some point, one of these studies need to include these basis uncertainties, at the very least to show that it doesn't make a difference (although I expect that it does).

We have followed the reviewer's suggestion to add greater historical context for the regression methodology employed in the study in Section 2.2 (see response to major point 1). We agree with the reviewer that there are limitations of multiple regression analysis, which we emphasise are not limited to examination of the SOR. However, it is difficult to conceive of other current approaches that would have significantly less limitations. We have added the following text to the manuscript at the end of Section 2.2: "It is a challenge in geophysical science to develop statistical methods to extract forced signals from complex timeseries. The implementation of multiple regression analysis as described above has a number of limitations, including (but not limited to): assumption that the input basis functions have zero uncertainty; difficulties in separating a signal from noise in a relatively short or sparse record (Damadeo et al 2014); and potential issues with degeneracy between basis functions (Chiodo et al 2014). These limitations should be kept in mind when examining detailed aspects of the results."

We inform the reviewer that there is now a dedicated working group within the SPARC SOLARIS-HEPPA activity that will perform a detailed comparison of statistical methods for analysing solar-climate signals with the eventual aim of providing some recommendations for best practices.

4) Anomalies. Often the authors use anomalies of variables, rather than the absolute variables. It would be good to see who real values of the data. I realise this can not always be done, but in some figures, for instance Figure 4, this would be very informative. Anomalies often make things look better!

Since the focus of our study is on quasi-decadal variability in ozone, we believe it makes sense to show anomalies from the long-term annual cycle, so that the vertical scale on the timeseries in Figures 2 and 4 can be sufficiently narrow that interannual to quasi-decadal variations are visible. However, to respond to the reviewer's request we have added figures to the Supplementary Material showing timeseries of absolute tropical ozone mixing ratios in the CCMI models (Figure S1) and in the climate model ozone databases (Figure S3).

5) There is a lot of focus on the CMIP6 ozone data set. But seems to be absolutely no citation to documentation on this data set. I note that the creators of the data set are not authors on this paper, and perhaps some of that lack of knowledge is reflected in the text. Is there a CMIP6 ozone paper coming out? Should this current paper be kept out of publication till that exists? This should certainly be true if there is any overlap.

Throughout this work we have liaised closely with the creators of the CMIP6 ozone database, led by Michaela Hegglin. We have sent Michaela the draft manuscript for

comment and she has even posted a comment on the discussion of this article in ACPD. At no point has it been indicated to us that our study should not be published. To the best of our knowledge the forthcoming publication in GMD describing the CMIP6 ozone database will not focus on the representation of the SOR, and thus we do not anticipate any significant overlap between the studies.

Two co-authors of our study (Dan Marsh and David Plummer) are the principal investigators of the CCMs (CMAM and CESM1(WACCM)) used to produce the CMIP6 ozone database. These co-authors have provided detailed information about the CCM simulations used to create the CMIP6 ozone database. The parts of this information that are particularly relevant to simulation of the SOR are described in Section 2.1.3 of the manuscript.

We remind the reviewer that the CMIP6 ozone database is publicly available and CMIP6 modellers are already implementing the dataset in their models: https://esgf-node.llnl.gov/projects/input4mips.

**Concerns (minor)**

1. Line 1: This alone would not fully capture the response 'fully capture' changed to 'to aid in capturing'

2. The SOR seems a little strange in this context, because it is not obvious (until later) that the SOR is not a 'set thing', it is only known within uncertainty bounds (and so different CCM give different SORs).

We use SOR throughout the manuscript for consistency with Part I. Here we have changed the text to say 'comparison of the representation of the solar-ozone response (SOR)....' to make clearer that this is something with variable representation across models.

3. Line 9: . . . ozone databases' – at this point it is not clear if the ozone data basis is the prescribed ozone, or simulated ozone from a CCMI.

Throughout the manuscript we distinguish between analysis of output from chemistryclimate models (in this case CCMI models), analysis of pre-calculated ozone databases for models without chemistry (which can be constructed from observations and/or CCMs), and analysis of ozone datasets (i.e. satellite observations taken from Part I). The use of 'database' in the manuscript is therefore solely reserved for pre-calculated ozone fields used in models without chemistry. We feel that the preceding sentence makes clear the distinction between the analysis of CCM results and of the pre-calculated ozone databases for CMIP5/CMIP6.

4. Line 11 Make clear that you refer to historical period ozone Time period of analysis added.

5. Line 13: weak compared with what? This clause has been removed.

6. Line 76 – a citation is really needed here (see major concern). The dataset is publicly available at: https://esgf-node.llnl.gov/projects/input4mips. This link has been added to the text.

7. Line 89 – what time frequency is this data? 'monthly mean' added

8. Line 94 'transferring' – please revise this word.

Text changed to 'may play a role in driving the 'top-down' mechanism for the solar cycle influence on high latitude regional surface climate (see e.g. Gray et al. (2010)).'

9. L124. Why only 1 ensemble member? Please repeat with all of them. You need to capture the uncertainty.

We have updated the analysis to use all available ensemble members for the models. See Table 1.

10. Page 5 (top): This seems very similar to Hood et al, please state that. We are unsure of what the reviewer is referring to as being similar to Hood et al (2015) and have therefore not changed the text.

11. Line 37: 5 x 5degree. This is not normal, why has the interpolation taken place? This is the resolution at which the SPARC/AC&C CMIP5 ozone database is provided: see Cionni et al. (2011) doi:10.5194/acp-11-11267-2011. This is because the historical part of the CMIP5 database was derived from satellite observations (SAGE I and II) that are available on a 5 degree grid. We have not performed any further interpolation.

12. Equ 1: Please cite where this came from originally.

Additional references have been added at the start of Section 2.2 that make reference to earlier work using similar methods, including the review of Staehelin et al (2001) that discuss the history of multiple regression methods for ozone trends.

13. Equ 1: Do the authors have any views on the breakdown between the long terms solar response, and the 11-year response?

We tested the sensitivity of our results to removing >11 year variability from the F10.7cm solar flux timeseries and found that removing the lower frequency solar variability had virtually no effect on the results. For simplicity we therefore did not perform any pre-filtering to the timeseries of the solar basis function.

14. Figure 1: Surely these QBO signals are just from one model? These will change. Yes, the QBO indices in Figure 1 are just an example based on the observed winds. This is now stated in the caption. The QBO indices for the models are calculated from the individual model wind fields as described in Section 2.2.

15. Line 218-219: 'better proxy' not convincing to me. Please cite a paper that compares these.

Floyd et al (2004, doi:10.1016/j.jastp.2004.07.013) show that F10.7cm and Mg-ii are correlated at >.95 for daily timeseries and >.99 for variability on timescales longer than several months. This reference has been added to the manuscript.

16. Line 220-225: This section on the volcanic signal is vague. The data sets the authors use are not long, in fact some figure just use  $\sim$ 30 years of data. Very short for regression. Volcanic signals will cause issues in the regression, and it is not clear the authors have dealt with the properly (nor in Part 1).

Following the reviewer's comment and after further discussions, in the revised manuscript we now adopt the approach of Maycock et al (2016) by removing data in the periods immediately following the three major tropical volcanic eruptions since 1960: Mt Agung, El Chichon and Mt Pinatubo. This is because the ozone response to volcanic eruptions is a non-linear function of chlorine amount and thus it is not appropriate to include a basis function for volcanic effects in the MLR model. The description in the Methods section has been updated to reflect this change.

17. Line 240. I think the authors need to show the autocorrelation plots to the reviewers, so we can assess this evidence. I agree they probably do not need to go in the main text.

Figure R2 below shows the e-folding time in months of the autocorrelation function (ACF) of the monthly regression residuals in the CCMI models. Areas where the e-folding time of the ACF is greater than 2 months are evident in all of the models in the mid and lower stratosphere and hence our choice to adopt an AR(2) model.

Figure R2: e-folding time [in months] of the autocorrelation function of the MLR residuals for each CCMI-1 model.

In testing the effects of the AR model choice, as requested by reviewer 1, we identified some sensitivity of the estimated SOR in the polar lowermost stratosphere, which may be related to the longer timescales of the ACF in that region in several models. The sensitivity to the AR model choice across the remainder of the stratosphere was small, which the exception of the tropical lower stratosphere in SOCOL, which is discussed in the revised text. Therefore to avoid giving potentially misleading information about the SOR in the polar lowermost stratosphere we have restricted the plots in the revised manuscript to a maximum pressure of 100 hPa.

18. Section 2.3: please describe more how this model fits into the wider models of CMIP5. Then we can assess suitability.

A detailed description of the model is given in Section 2.3 of the manuscript. The model has a well-resolved stratosphere (model lid height above 50 km) and simulates the major aspects of the stratospheric circulation e.g. sudden warmings, the QBO (see e.g. Charlton-Perez et al., 2013; Schmidt et al., 2013). The response to 11 year solar forcing

in the CMIP5 version of ECHAM has been shown to be comparable to other high-top stratosphere resolving CMIP5 models (Mitchell et al., 2015).

Since the model does not include interactive chemistry, it provides a suitable test-bed for quantifying the effects of the pre-calculated ozone databases for CMIP5 and CMIP6.

19. Line 255-260: This is worrying that the lower signal might not be so well captured. We are unsure what the reviewer means by 'lower signal' in this context. The impact of the short wavelength absorption below 200 nm that is not captured in the ECHAM6.3 radiative code is quantified by Sudhokolov et al (2014) – see lower right panel of their Figure 2. The underestimation of the solar max-min shortwave heating anomaly in the stratosphere is ~15-20%, but is much larger above 60 km and thus we restrict our analysis to the stratosphere (

---

## Author Response (AR2)

Author response to second review Anonymous Reviewer 2

Overall the Authors have done many corrections on their paper, and have addressed most of my minor concerns satisfactorily. Some of the responses to major concerns are not satisfactory, especially regarding the novelty of the analysis. My overall conclusion is that while I do not necessarily agree with some of the implementations of methods, there is nothing fundamentally wrong with the analysis either (it is just not as rigorous as it could be). But, the paper provides only a small incremental advance to the field, and is not what I would call novel. As such, I do not believe it warrants publication in this journal.

I will not go through all the major point responses from the author, only the one regarding novelty, which the authors state is summarised in the introductory section with 4 points. I will take the 4 points separately, and explain why I believe the paper is only an incremental advance.

We respectfully disagree with the reviewer's resolute view that the manuscript is not sufficiently novel for publication in Atmospheric Chemistry and Physics. Our justification for this position is given below in blue.

1. To provide an update to previous CCM studies by analysing the SOR in CCMI-1 models.

Hardly ground breaking science. All that has been done here is using very similar methods to very similar data, and unsurprisingly getting very similar results. The previous similar studies were only a couple of years ago – models haven't advanced that much in two years.

The solar-ozone response in CCMs (CCMVal-1) were first analysed more than 10 years by Austin et al. (2008). CCMVal-2 results were briefly described in SPARC CCMVal-2 (2010) report but were never published.  Hood et al. (2015) analysed a very limited subset of CMIP5 models (3 independent models that included a realistic representation of solar variability). The progress in the general performance of CCMs is evident if we compare CCMVal-1 and CCMI for other measures such as long-term ozone trends (e.g. Dhomse et al., 2018). Therefore, it is important and new to document the solar component of their performance.  Getting the same results is not a problem. It just confirms that the solar response is robust. It is new and should be done, because otherwise one could not say whether or not the response to solar variability in the models is the same a decade after Austin et al. (2008). We have amended the Introduction to make this point clearer.

2. To evaluate the SOR in three pre-calculated ozone databases for climate models from CMIP5, CMIP6 and Bodecker et al.

Yes, but this analysis has already been performed for two out of these three datasets, so this is wildly overstating the novelty. It is new for CMIP6, but again, see my response to point 1....

The extracted solar-ozone coefficients from CMIP6 are a valuable dataset for other modeling projects. For example, the coefficients described in the study have already been used in PMIP4 in order to have a consistent representation of the solar-ozone response between PMIP4 and CMIP6 (Jungclaus et al., 2017; Bader et al., in preparation). It is therefore important that this paper documents the CMIP6 solar-ozone coefficients that are being used by other parts of the community. The coefficients are now being published along with the manuscript as a dataset (https://doi.org/10.5518/348), which is available to other researchers. The comparison with CMIP5 simply helps to put the latest dataset into the context of previous work. The citation to the ozone coefficients dataset has been added to the manuscript.

3. To compare CCMs and ozone databases with satellite observations from Part 1.

But this is exactly what Hood et al, 2015 did. They compared satellites with CCMs, and with nonchemistry models. Clearly seen in their figures and analysis. As Hood is an author on this paper, perhaps Maycock and Hood can discuss this and present an answer. At the moment, I see very little novelty (other than, yet again, the application to slightly different data sets, but you've already covered that in points 1 and 2).

Hood et al. (2015) did not include the latest satellite datasets. They used SBUV VN8.0 and SAGE II VN6.2; the latter dataset has been shown to have significant limitations in its representation of solar variability (Maycock et al., 2016; Dhomse et al., 2016). We use the latest version SAGE VN7.0 extended to the recent past for comparison with the models. This point has been added in the Introduction. If the results are similar to previous findings it is not so important, because we did not know this would be the case a priori.

4. To perform atmospheric model experiments….

This is certainly the most novel aspect of the paper, and in my view by far the most useful part. Model responses to the simulated atmospheric response (even with the same ozone prescribed) are widely different though, so for this to be a robust analysis, the experiments should be performed in at least one other model.

We show that in a CMIP5/6 model that the changes in solar ozone response have a first order effect on the simulated temperature response to the 11 year solar cycle. This result could vary slightly from model to model, but basic radiative transfer says that imposing a solar-ozone response that is almost half the amplitude in the upper tropical stratosphere will result in a smaller temperature response, so there is no reason to suspect that the model is inconsistent with that effect. The CMIP6 solar ozone coefficients are now being published with the manuscript (see point 2), and are therefore available to other groups to perform their own model experiments to check this result.

**References**

Austin, J., K. Tourpali, E. Rozanov, H. Akiyoshi, S. Bekki, G. Bodeker, C. Brühl, N. Butchart, M. Chipperfield, M. Deushi, V. I. Fomichev, M. A. Giorgetta, L. Gray, K. Kodera, F. Lott, E. Manzini, D. Marsh, K. Matthes, T. Nagashima, K. Shibata, R. S. Stolarski, H. Struthers, and W. Tian, Coupled chemistry climate model simulations of the solar cycle in ozone and temperature, J. Geophys. Res., 113, D11306, 2008.

Dhomse, S. S., M. P. Chipperfield, R. P. Damadeo, J. M. Zawodny, W. T. Ball, W. Feng, R. Hossaini, G. W. Mann, and J. D. Haigh (2016), On the ambiguous nature of the 11 year solar cycle signal in upper stratospheric ozone, Geophys. Res. Lett., 43, 7241–7249, doi:10.1002/2016GL069958.

Dhomse, S., Kinnison, D., Chipperfield, M. P., Cionni, I., Hegglin, M., Abraham, N. L., Akiyoshi, H., Archibald, A. T., Bednarz, E. M., Bekki, S., Braesicke, P., Butchart, N., Dameris, M., Deushi, M., Frith, S., Hardiman, S. C., Hassler, B., Horowitz, L. W., Hu, R.-M., Jöckel, P., Josse, B., Kirner, O., Kremser, S., Langematz, U., Lewis, J., Marchand, M., Lin, M., Mancini, E., Marécal, V., Michou, M., Morgenstern, O., O'Connor, F. M., Oman, L., Pitari, G., Plummer, D. A., Pyle, J. A., Revell, L. E., Rozanov, E., Schofield, R., Stenke, A., Stone, K., Sudo, K., Tilmes, S., Visioni, D., Yamashita, Y., and Zeng, G.: Estimates of Ozone Return Dates from Chemistry-Climate Model Initiative Simulations, Atmos. Chem. Phys. Discuss., https://doi.org/10.5194/acp-2018-87, in review, 2018.

Jungclaus, J. H., Bard, E., Baroni, M., Braconnot, P., Cao, J., Chini, L. P., Egorova, T., Evans, M., González-Rouco, J. F., Goosse, H., Hurtt, G. C., Joos, F., Kaplan, J. O., Khodri, M., Klein Goldewijk, K., Krivova, N., LeGrande, A. N., Lorenz, S. J., Luterbacher, J., Man, W., Maycock, A. C., Meinshausen, M., Moberg, A., Muscheler, R., Nehrbass-Ahles, C., Otto-Bliesner, B. I., Phipps, S. J., Pongratz, J., Rozanov, E., Schmidt, G. A., Schmidt, H., Schmutz, W., Schurer, A., Shapiro, A. I., Sigl, M., Smerdon, J. E., Solanki, S. K., Timmreck, C., Toohey, M., Usoskin, I. G., Wagner, S., Wu, C.-J., Yeo, K. L., Zanchettin, D., Zhang, Q., and Zorita, E.: The

PMIP4 contribution to CMIP6 – Part 3: The last millennium, scientific objective, and experimental design for the PMIP4 past1000 simulations, Geosci. Model Dev., 10, 4005-4033, https://doi.org/10.5194/gmd-10-4005-2017, 2017.